# Improved high-dimensional estimation with Langevin dynamics and stochastic weight averaging

**Stanley Wei**
Princeton University

**Alex Damian**
Harvard University

**Jason D. Lee**
UC Berkeley

## Abstract

Significant recent work has studied the ability of gradient descent to recover a hidden planted direction $\theta^\star \in S^{d-1}$ in different high-dimensional settings, including tensor PCA and single-index models. The key quantity that governs the ability of gradient descent to traverse these landscapes is the *information exponent* $k^\star$ (Ben Arous et al., 2021), which corresponds to the order of the saddle at initialization in the population landscape. Ben Arous et al. (2021) showed that $n \gtrsim d^{\max(1,k^\star-1)}$ samples were necessary and sufficient for online SGD to recover $\theta^\star$, and Ben Arous et al. (2020) proved a similar lower bound for Langevin dynamics. More recently, Damian et al. (2023) showed it was possible to circumvent these lower bounds by running gradient descent on a smoothed landscape, and that this algorithm succeeds with $n \gtrsim d^{\max(1,k^\star/2)}$ samples, which is optimal in the worst case. This raises the question of whether it is possible to achieve the same rate *without explicit smoothing*. In this paper, we show that Langevin dynamics can succeed with $n \gtrsim d^{k^\star/2}$ samples if one considers the *average iterate*, rather than the last iterate. The key idea is that the combination of noise-injection and iterate averaging is able to emulate the effect of landscape smoothing. We apply this result to both the tensor PCA and single-index model settings. Finally, we conjecture that minibatch SGD can also achieve the same rate without adding any additional noise.

## 1 Introduction

In many learning settings, gradient descent is the default algorithm, and recent years have seen significant progress in understanding its theoretical properties and learnability guarantees in different feature learning settings (Damian et al., 2022; Mei et al., 2022). While the optimization process is non-convex in general, there are many settings in which we can nonetheless tractably give learning guarantees. Single index models, or functions of the form $\sigma(\theta^\star \cdot x)$, provide one such sandbox; here, the goal is to recover this planted direction $\theta^\star \in S^{d-1}$ through which the target depends on the input. In the statistics literature, single index models have been studied for decades (Hristache et al., 2001; Härdle et al., 2004), and are also known as generalized linear models. In the special case where the link function $\sigma$ is monotonic, the information-theoretic sample complexity of $n \asymp d$ to learn $\theta^\star$ is achieved via perceptron-like algorithms (Kalai and Sastry, 2009; Kakade et al., 2011). For non-monotonic link functions, one classic example is the phase-retrieval problem where $\sigma(t) = |t|$, which has been well-studied (Chen et al., 2019; Maillard et al., 2020).

For the case of Gaussian input data, the information exponent $k^\star$ of the link function $\sigma$ tells us the sample complexity needed to learn $\theta^\star$ with "correlational learners" (Ben Arous et al., 2021). This can be extended to allow for label preprocessing (Mondelli and Montanari, 2018; Maillard et al., 2020; Chen et al., 2025; Dandi et al., 2024; Troiani et al., 2024; Lee et al., 2024; Arnaboldi et al., 2024) and the resulting exponent becomes the "generative exponent" (Damian et al., 2024). Ben Arous et al. (2021) shows that using $n \gtrsim d^{k^\star-1}$ samples is necessary and sufficient for a certain class of online stochastic gradient descent (SGD) algorithms. Damian et al. (2023) improves this to $n \gtrsim d^{\max(1,k^\star/2)}$ samples by running online SGD on a smoothed loss, and they provide a matching correlational statistical query (CSQ) lower bound. Key to their analysis is the fact that the

smoothed loss boosts the signal-to-noise ratio in the region near initialization (i.e. when the current iterate lies in the equatorial region with respect to $\theta^\star$).

Overall, the information exponent has been shown to determine the sample complexity in many settings (Ben Arous et al., 2021; Damian et al., 2023; Bietti et al., 2022; Abbe et al., 2023; Dandi et al., 2023). A recent work of Joshi et al. (2025) analyzes the spherical symmetric distribution case, which slightly relaxes the Gaussian data assumption. In particular, the work by Abbe et al. (2023) provides a generalization of the information exponent to the multi index setting, in which the target depends on a low dimensional subspace of the input instead of just a single direction (Ren and Lee, 2024; Damian et al., 2025). We would also like to note the connection of learning information exponent $k$ single index models to the order $k$ tensor PCA problem (Montanari and Richard, 2014). In both problems, it turns out that the partial trace estimator returns the direction of the planted spike with optimal sample complexity of $d^{k/2}$ in the CSQ framework, and similar smoothing-based approaches there (Anandkumar et al., 2017; Biroli et al., 2020) have been proposed to return this estimator.

Notably, along this line of work, Ben Arous et al. (2020) conjectures that Langevin dynamics in the tensor PCA setting does not work due to the divergence of the computational-statistical gap in this setting. In our work, we *surprisingly* show that Langevin dynamics can still be used to recover the planted direction of the single index model. To achieve this, we run Langevin dynamics, but we take the *time average* of all the iterates. Our analysis reveals that with $n \gtrsim d^{\lceil k^\star/2 \rceil}$ samples, we are able to recover the direction of the partial trace estimator and hence $\theta^\star$. The key insight is that this Langevin dynamics process closely tracks the Brownian motion on the sphere, and averaging out the iterates roughly corresponds to an ergodicity concentration argument on the sphere. Our main theorem is the following.

**Theorem 1** (Main theorem (informal))**.** *Consider a link function $\sigma$ with information exponent $k^\star$. Then, with $n \gtrsim d^{\lceil k^\star/2 \rceil}$ samples drawn i.i.d. from the standard $d$-dimensional Gaussian, running Algorithm 1 recovers the ground truth direction $\theta^\star$.*

We can also shave off a factor of $\sqrt{d}$ to improve the sample complexity to $n \gtrsim d^{k^\star/2}$ by running Algorithm 1 and running online SGD on the returned time averaged estimator. This corresponds to the warm start in Damian et al. (2023) for the odd case.

## 2 Setup and Main Contributions

### 2.1 Notation

We use $\|\cdot\|_p$ to denote the vector $\ell_p$-norm; furthermore, when $p = 2$, we often drop the subscript and write $\|\cdot\|$. Given a probability measure $\gamma$ over $\mathbb{R}^d$, we denote $L^2(\mathbb{R}^d, \gamma)$ the space of $\gamma$-measurable and square-integral functions; we shorthand this to $L^2(\gamma)$ when the domain is clear. For $f \in L^2(\gamma)$, we denote $\|f\|_{L^2(\gamma)}^2 = \mathbb{E}_{z\sim\gamma}[f(z)^2]$. We also denote $\mu$ to be the uniform measure on $S^{d-1}$.

### 2.2 Setting

We consider in this paper tensor PCA (Montanari and Richard, 2014) and single-index models.

#### 2.2.1 Tensor PCA

For tensor PCA, we will assume there is a planted direction $\theta^\star \in S^{d-1}$ and we observe the $k$-tensor $T$ defined by:

$$T = \theta^{\star\otimes k} + n^{-1/2}Z \quad \text{where} \quad Z_{i_1,\ldots,i_k} \overset{\text{i.i.d.}}{\sim} N(0,1)$$

We consider optimizing the negative log-likelihood:

$$L(\theta) = -\left\langle T, \theta^{\otimes k} \right\rangle$$

Information theoretically, $\theta^\star$ is possible to recover whenever $n \gtrsim d$. However, common techniques like approximate message passing (AMP), tensor power method, and online SGD require $n \gtrsim d^{k-1}$

to recover $\theta^\star$ (Montanari and Richard, 2014; Ben Arous et al., 2021). Nevertheless, it is possible to recover $\theta^\star$ with $n \gtrsim d^{k/2}$ samples using tensor unfolding (Montanari and Richard, 2014), the partial-trace estimator (Hopkins et al., 2016), and landscape smoothing (Anandkumar et al., 2017; Biroli et al., 2020; Damian et al., 2023). In our paper, we show Langevin dynamics combined with iterate averaging can recover $\theta^\star$ with $n \gtrsim d^{\lceil \frac{k}{2} \rceil}$ without explicit unfolding or smoothing.

### 2.2.2 SINGLE-INDEX MODELS

We mostly follow the setting of Damian et al. (2023). Let $\{(x_i, y_i) \in \mathbb{R}^d \times \mathbb{R}\}_{i \in [n]}$ be the set of training data. The input data $x_i$ are drawn i.i.d. from a standard $d$-dimensional Gaussian $\mathcal{N}(0, I_d)$, and the labels $y_i$ are generated through a target or teacher function $f^\star$. In particular, we consider the setting where $f^\star$ is a single index model, in which the label only depends on the input through a planted direction $\theta^\star \in S^{d-1}$. Formally, we have for each $i$:

$$y_i = f^\star(x_i) + \xi_i = \sigma(\theta^\star \cdot x_i) + \xi_i, \quad x_i \overset{\text{i.i.d.}}{\sim} \mathcal{N}(0, I_d), \xi_i \overset{\text{i.i.d.}}{\sim} \mathcal{N}(0, 1)$$

where $\sigma$ is a known link function. We will consider the setting where our learner is $f(\theta, x) := \sigma(\theta \cdot x)$, where $\theta \in S^{d-1}$ is the learnable parameter.

**Assumption 1.** *We will assume the following regarding the link function $\sigma$.*

- $\mathbb{E}_{x \sim \mathcal{N}(0,1)}[\sigma(x)^2] = 1$ *(Normalization)*

- $|\sigma^{(k)}(z)| \leq C$ *for $k = 0, 1, 2$ and for all $z$. (Lipschitzness)*

We note the assumption on the boundedness of $\sigma^{(k)}$ can be relaxed to it having polynomial tails Damian et al. (2023), but at the cost of increasing the complexity of the proof.

We consider training via the correlation loss; the loss on a specific sample $(x, y)$ is:

$$L(\theta; x, y) = 1 - f(\theta, x)y$$

The empirical loss on our training set is therefore:

$$L_n(\theta) = \frac{1}{n} \sum_{i \in [n]} L(\theta; x_i, y_i)$$

We also denote the population loss over $(x, y)$ from the data distribution to be $L(\theta) := \mathbb{E}_{(x,y)}[L(\theta; x, y)]$.

In this setting, Ben Arous et al. (2021) showed that the sample complexity for learning depends on a quantity called the information exponent $k^\star$ of the link function $\sigma$. To motivate this definition, consider first the probabilist's Hermite polynomials.

**Definition 1** (Probabilist's Hermite polynomials). *For $k \geq 0$, the $k$th normalized probabilist Hermite polynomial $h_k : \mathbb{R} \to \mathbb{R}$ is:*

$$h_k(x) = \frac{(-1)^k}{\sqrt{k!}} \gamma(x)^{-1} \frac{d^k}{dx^k} \gamma(x)$$

*where $\gamma(x) := \frac{e^{-x^2/2}}{\sqrt{2\pi}}$ is the probability density function of a standard univariate Gaussian.*

Of importance is that the Hermite polynomials form an orthogonal basis in $L^2(\gamma)$ (i.e. the space of square-integrable functions with respect to the standard Gaussian measure). Henceforth, for link function $\sigma \in L^2(\gamma)$, let $\{c_k\}_{k \geq 0}$ denote the Hermite coefficients of $\sigma$:

**Definition 2** (Hermite coefficients). *Let the Hermite coefficients of $\sigma \in L^2(\gamma)$ be $\{c_k\}_{k \geq 0}$. In other words,*

$$\sigma(x) = \sum_{k=0}^{\infty} c_k h_k(x), \quad c_k = \mathbb{E}_{z \sim \mathcal{N}(0,1)}[\sigma(z)h_k(z)]$$

This leads us to the key quantity, the information exponent.

**Definition 3** (Information exponent). *We define the information exponent to be:*

$$k^\star = \min\{k \geq 1 : c_k \neq 0\}$$

In other words, this is the first Hermite coefficient with positive index that is nonzero. Some examples of information exponents are below:

**Example 1.** *(Link functions and their information exponents)*

- $\sigma(t) = t$ and $\sigma(t) = \text{ReLU}(t)$ *have information exponent 1.*

- $\sigma(t) = |t|$ and $\sigma(t) = t^2$ *have information exponent 2.*

- $\sigma(t) = t^2 e^{-t^2}$ *has information exponent 4.*

- $\sigma(t) = h_k(t)$ *has information exponent $k$.*

Ben Arous et al. (2021) showed that $n \gtrsim d^{\max(1, k^\star - 1)}$ samples were necessary and sufficient for online SGD to recover $\theta^\star$, mirroring the tensor PCA setting. Damian et al. (2023) showed that this rate could be improved to $n \gtrsim d^{\max(1, k^\star/2)}$ by running online SGD on a smoothed landscape. A number of papers have managed to circumvent the information exponent by applying a label transformation before running SGD Mondelli and Montanari (2018); Maillard et al. (2020); Chen et al. (2025); Dandi et al. (2024); Troiani et al. (2024); Damian et al. (2024); Lee et al. (2024). These results apply a transformation $\mathcal{T}$ to the labels $\{y_i\}_{i=1}^n$ to derive samples from the single index model defined by $\mathcal{T} \circ \sigma$. This link function can have smaller information exponent than $\sigma$, and the smallest exponent such a transformation can achieve is called the "generative exponent" Damian et al. (2024). For the purposes of this paper, we can assume that such a label transformation has already been applied so that the information exponent and the generative exponent coincide.

## 2.3 THE LEARNING ALGORITHM

**Definition 4** (Spherical gradient operator). *For $\theta \in S^{d-1}$ and function $g : \mathbb{R}^d \to \mathbb{R}$, define the spherical gradient operator to be $\nabla_\theta g(\theta) = P_z^\perp \nabla g(z)|_{z=\theta}$, where $P_\theta^\perp := I - \frac{\theta\theta^\top}{\|\theta\|^2}$ is the orthogonal projection operator with respect to $\theta$ and $\nabla$ is the standard Euclidean gradient.*

We now formally define our learning algorithm; here, $\{W_t\}_{t \geq 0}$ is the standard Wiener process in $\mathbb{R}^d$.

---

**Algorithm 1** Learning algorithm

---

**Input:** Inverse temperature parameter $\epsilon$, number of time steps $T$, data points $\{(x_i, y_i)\}_{i=1}^n$
Initialize $\theta_0 \sim \mu$ (e.g. uniform over $S^{d-1}$)
Run the following SDE up to time $T$:

$$d\theta = \left(-\frac{d-1}{2}\theta + \epsilon b(\theta)\right)dt + P_\theta^\perp dW_t, \quad b(\theta) := -\nabla_\theta L_n(\theta) \tag{1}$$

$\hat{\theta} := \frac{1}{T}\int_0^T \theta_t dt$
$\hat{M} := \frac{1}{T}\int_0^T \theta_t \theta_t^\top dt$
**If $k^\star$ is odd**, return $\hat{\theta}/\|\hat{\theta}\|$
**Otherwise if $k^\star$ is even**, return the top eigenvector $v_1$ of $\hat{M}$

---

It can be shown that when $\theta_t$ follows the SDE in Equation (1), it remains on the sphere for all time $t$. Thus, this SDE is the natural analogue of the standard Langevin dynamics on the sphere. A discussion regarding this is deferred to the appendix.

## 2.4 MAIN CONTRIBUTIONS

We now highlight our main contributions in this work.

- We show that by combining Langevin dynamics with weight averaging, we can recover $\theta^\star$ in both the tensor PCA and single-index model settings with $n \gtrsim d^{\lceil k^\star/2 \rceil}$ samples, which nearly matches the optimal computational-statistical tradeoff for these problems (Damian et al., 2024; Hopkins et al., 2015).

- In contrast with previous work (Damian et al., 2023; Biroli et al., 2020; Anandkumar et al., 2017), which attain the sample complexity guarantee via smoothing the existing loss landscape to create a high signal-to-noise ratio regime, we utilize the other end of the spectrum - a low signal-to-noise ratio setting. Our method of uniform averaging takes advantage of the noise, and allows us to learn the estimator that one would obtain by running landscape smoothing.

- One other feature of our algorithm is that it does not see the data in an online manner, unlike previous works (Damian et al., 2023; Ben Arous et al., 2021). We use the empirical risk minimization (ERM) loss to obtain our results.

- (Ben Arous et al., 2020) shows that Langevin dynamics struggles to escape the "equator" $\{\theta \ : \ |\theta \cdot \theta^\star| \lesssim d^{-1/2}\}$ without $n \gtrsim d^{k^\star-1}$ samples. Surprisingly, we show that it is not necessary to escape the equator to get a good estimate of $\theta^\star$ – our process $\theta(t)$ indeed lies on the equator throughout the training process so that its correlation with $\theta^\star$ remains small, but the *time-averaged iterate* can still converge to $\theta^\star$.

## 3 MAIN RESULTS

Our high level framework is to show ergodic concentration to an estimator that recovers the planted direction with enough samples. We will state our results for both the odd and even algorithm.

**Theorem 2** (Odd $k^\star$). *Let $\epsilon = o\big(d^{-(k^\star-3)/2}\big)$ and $T \gtrsim d^{k^\star}/\epsilon^2$. Then, Algorithm 1 succeeds in estimating $\frac{2\epsilon}{d-1}\mathbb{E}_{z\sim\mu}[b(z)]$ up to $O(\epsilon)$ relative error. Moreover, for $\Delta > 0$, if $n \gtrsim d^{\lceil k^\star/2 \rceil}/\Delta^2$, we recover the ground truth $\theta^\star$ up to error $\Delta$ with probability at least $1 - e^{d^c}$.*

Consider first the setting where $\epsilon \to 0$; this corresponds to a convergence to the pure Brownian motion on $S^{d-1}$, which has Itô SDE

$$d\beta = \left(-\frac{d-1}{2}\beta\right)dt + P_\beta^\perp dW_t$$

In the regime of $\epsilon$ in Theorem 2, it turns out that at time $t$, we can write $\theta_t = \beta_t + E_t$ where $E_t$ is an error term of order $\epsilon$, and we couple the processes $\theta$ and $\beta$ with the same noise process $W_t$. We set $\theta_0 = \beta_0$, and $E_0 = 0$, with the former being drawn from the uniform distribution on the sphere. Then, time averaging allows us to obtain:

$$\frac{1}{T}\int_0^T \theta_t dt = \frac{1}{T}\int_0^T \beta_t dt + \frac{1}{T}\int_0^T E_t dt$$

By ergodicity of Brownian motion, we can prove that the first term concentrates to zero. For the second term $E_t$, we show that the time average of it converges to the direction of $\mathbb{E}_{z\sim\mu}[\nabla L_n(z)]$. In both the tensor PCA and single-index model settings, this estimator can be shown to recover the planted direction $\theta^\star$ with $n \gtrsim d^{\lceil k^\star/2 \rceil}$ samples. Moreover, it is possible to use this estimator as a warm start before running online SGD. This idea was also used by Hopkins et al. (2016); Anandkumar et al. (2017); Damian et al. (2023) to boost this estimator, and allow it to recover $\theta^\star$ with $n \gtrsim d^{k^\star/2}$ samples:

**Corollary 1.** *Using the same $\epsilon$ and $T$ in the setting of Theorem 2 and $n = \Omega(d^{k^\star/2})$, we can run Algorithm 1, followed by online SGD with $\Omega(d^{k^\star/2})$ samples to recover the ground truth $\theta^\star$ to arbitrary accuracy.*

The idea here is with $n = \Omega(d^{k^\star/2})$ samples (which is a multiple of $\sqrt{d}$ less than in Theorem 2), the averaging estimator gives us a warm start that obtains correlation $\Theta(d^{-1/4})$ with $\theta^\star$. From here, we can run online SGD using the result from Ben Arous et al. (2021) to recover the ground truth. We now proceed to state our result for the even case.

**Theorem 3** (Even $k^\star$). *Let $\epsilon = o(d^{-(k^\star-2)/2})$, and let $T \gtrsim d^{k^\star+1}/\epsilon^2$. Then, Algorithm 1 succeeds in estimating $\mathbb{E}_{z\sim\mu}[zz^\top] + \frac{\epsilon}{d}\mathbb{E}_{z\sim\mu}[zb(z)^\top + b(z)z^\top]$ up to $O(\epsilon)$ relative error in operator norm. Moreover, for $\Delta > 0$, if $n \gtrsim d^{k^\star/2}/\Delta^2$, then the top eigenvector of our estimator recovers the ground truth $\theta^\star$ up to error $\Delta$ with probability at least $1 - e^{d^c}$.*

Intuitively, the algorithm for the odd case does not work here because of the first order terms vanish upon taking time average, due to the symmetry of the uniform distribution/Brownian motion. More specifically, $\mathbb{E}_{z\sim\mu}[\nabla L_n(z)] \approx 0$ and does not have any meaningful correlation with $\theta^\star$. On the other hand, when we consider the time average of the second order information given by $\theta\theta^\top$, we can precisely recover the planted direction $\theta^\star$ by taking the top eigendirection of our estimator. More formally, time averaging gives us:

$$\frac{1}{T}\int_0^T \theta_t\theta_t^\top dt = \frac{1}{T}\int_0^T \beta_t\beta_t dt + \frac{1}{T}\int_0^T (\beta_t E_t^\top + E_t\beta_t^\top)dt + \frac{1}{T}\int_0^T E_t E_t^\top$$

We prove concentration of each of these terms to the stationary average via the ergodicity of the spherical Brownian motion, which leads to a final quantity of approximately $\mathbb{E}_{z\sim\mu}[zz^\top] + \frac{\epsilon}{d}\mathbb{E}_{z\sim\mu}[zb(z)^\top + b(z)z^\top]$. The first term converges to $I/d$, and the final term is a negligible error term. When $n \gtrsim d^{k^\star/2}$, the middle term converges to a matrix with a rank-one spike $\theta^\star\theta^{\star\top}$.

## 4 OVERVIEW OF PROOF IDEAS

### 4.1 ERGODIC CONCENTRATION

In showing a general ergodic concentration result, we first give some preliminaries on Markov processes on compact Riemannian manifolds.

**Definition 5** (Markov semigroup). *Let $(X_t)_{t\geq 0}$ be a time-homogeneous Markov process. Then, its associated Markov semigroup $(P_t)_{t\geq 0}$ is the family of operators acting on bounded measurable functions $f$ through:*

$$P_t f(x) := \mathbb{E}[f(X_t)|X_0 = x]$$

At this point, it is useful to define the infinitesimal generator of a Markov process.

**Definition 6** (Infinitesimal generator). *Let $(P_t)_{t\geq 0}$ be the associated Markov semigroup for a Markov process. Then, the infinitesimal generator $\mathcal{L}$ associated with this semigroup is defined as:*

$$\mathcal{L}f := \lim_{t\to 0}\frac{P_t f - f}{t}$$

*for all functions $f$ for which this limit exists.*

Having these definitions introduced, consider the Brownian motion on $S^{d-1}$ that we defined earlier:

$$d\beta = \left(-\frac{d-1}{2}\beta\right)dt + P_\beta^\perp dW_t$$

Note that by rotational invariance, the stationary distribution is $\mu$. Moreover, by classic results (Saloff-Coste, 1994), we know that the infinitesimal generator of this process is $\mathcal{L} = \frac{1}{2}\Delta_{S^{d-1}}$, where $\Delta_{S^{d-1}}$ is the Laplace-Beltrami operator on $S^{d-1}$. We now give a general lemma for ergodic averages of functions of a Brownian motion over the sphere.

**Lemma 1.** *Let $f : \mathbb{R}^d \to \mathbb{R}$ such that $f \in L^2(\mu)$, where $\mu$ is the stationary uniform measure over the sphere for the Brownian motion, and $\int_{S^{d-1}} f d\mu = 0$. Then, we have:*

$$\frac{1}{T}\int_0^T f(\beta_t)dt = \frac{\phi(\beta_0) - \phi(\beta_T)}{T} + \frac{M_T}{T}$$

*where*

$$\phi(\beta) = \int_0^\infty P_t f(\beta)dt$$

*and $M_T := \int_0^T \nabla\phi(\beta_t)^\top P_{\beta_t}^\perp dW_t$ is a martingale.*

The proof is deferred to the appendix, and it now remains to bound these terms, which depends on our choice of $f$. Recall that we need to make this ergodicity argument for $\beta_t$ and $b(\beta_t)$ (defined in Section 4.2), as well as $\beta_t b(\beta_t)^\top$ for the even case. For the sake of exposition, we look at this function coordinate-wise in the main text; our full proofs in the appendix directly handle the tensorized version.

The bounds on these quantities are given by the following lemma, with full proof in the appendix.

**Lemma 2.** *In the setting of Lemma 1 the following holds:*

$$\left\| \frac{\phi(\beta_0) - \phi(\beta_T)}{T} \right\| \le \frac{2 \sup \|\nabla f\|}{(d-2)T}$$

$$\mathbb{E}\left[ \left( \frac{M_T}{T} \right)^2 \right] \le \frac{\sup \|\nabla f\|^2}{(d-2)^2 T}$$

The $d - 2$ term comes from the Ricci curvature of $S^{d-1}$ being $\rho = d - 2$, which leads to a bound on the gradient decay in the sense that $\|\nabla P_t f\| \le e^{-\rho t}\|\nabla f\|$ (Bakry et al., 2016). A detailed discussion of this is included in the appendix. We now sketch the remainder of the ergodicity arguments in the main result. The previous lemmas tell us that the concentration happens at time $T$ that depends on the function $f$.

## 4.2 Analyzing the Error Component $E$

Recall in the previous section that the time average consists of a Brownian component that is averaged out to zero, and an error component $\frac{1}{T} \int_0^T E_t dt$. First, let us recall our definition $b(\theta) := -\nabla_\theta L_n(\theta) = \frac{1}{n} P_\theta^\perp \sum_{i \in [n]} y_i \sigma'(\theta \cdot x_i) x_i$. By decomposing the time average of $E_t$ even further, it turns out we can write the above as roughly:

$$\frac{1}{T} \int_0^T E_t dt \approx \frac{\epsilon}{d} \frac{1}{T} \int_0^T b(\theta_t) dt$$

From here, we derive the following:

$$\frac{1}{T} \int_0^T b(\theta_t) dt = \frac{1}{T} \int_0^T b(\beta_t) dt + \frac{1}{T} \int_0^T (b(\theta_t) - b(\beta_t)) dt$$

The first term concentrates to $\bar{b} := \mathbb{E}_{z \sim \mu}[b(z)]$ using the ergodicity arguments from the previous section, and the second term can be controlled via upper bound on $\|E_t\| = \|\theta_t - \beta_t\|$ due to Lipschitzness. Indeed, in the regime of $\epsilon$ that we work in, we can further argue that with high probability, $\|\theta - \beta\|$ remains order $O(\epsilon)$ over all time, which we outline below. Recall the SDE's for the coupled processes $\theta, \beta$:

$$d\theta = \left( -\frac{d-1}{2}\theta + \epsilon b(\theta) \right) dt + P_\theta^\perp dW_t$$

$$d\beta = -\frac{d-1}{2}\beta dt + P_\beta^\perp dW_t$$

This tells us that:

$$dE = \left( -\frac{d-1}{2}E + \epsilon b(\theta) \right) dt + \left( P_\theta^\perp - P_\beta^\perp \right) dW_t$$

The key observation here is that the noise matrix $\Sigma^{1/2} := P_\theta^\perp - P_\beta^\perp$ satisfies the property that $\operatorname{tr} \Sigma \le 2\|E\|^2$. Intuitively, this means that the size of the noise scales with the norm of $E$, and this allows us to get a high probability uniform bound on $\|E\|$ over all time. The following lemma makes this rigorous.

**Lemma 3** (High probability uniform bound of $\sup \|E\|$). *With probability at least $1 - dTe^{-d}$, there exists an absolute constant $C'$ such that:*

$$\sup_{t \le T} \|E(t)\| \le C' \left[ \frac{\epsilon \sup \|b\|}{d} \right]$$

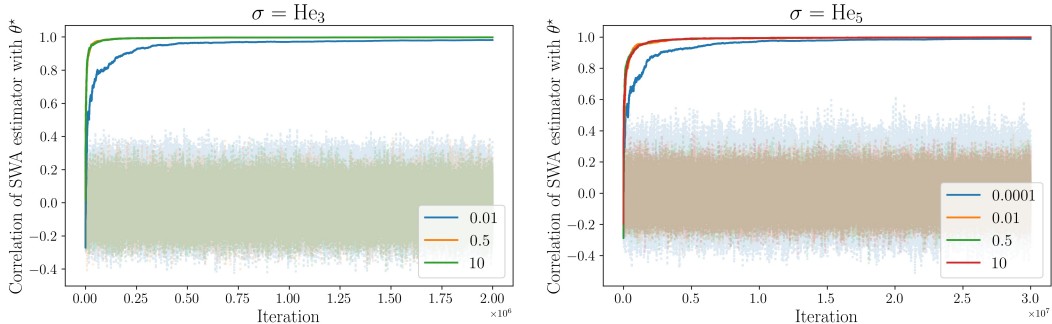

Figure 1: We run with $d = 100$ with $n = 10d^{\lceil k^{\star}/2 \rceil}$ samples, using various learning rates. Here, the dark curves correspond to the correlation of the time average as a function of iteration, in which it indeed converges to the direction of $\theta^{\star}$. The light curves correspond to the actual iterate as a function of time, which can be seen to stay near the equator over the entire training process.

The key idea of this uniform bound lies in a bijection between this Ornstein–Uhlenbeck-like process and a suitable subgaussian process. From there, we can apply the chaining method to obtain a uniform bound of $\sup \|E\|$ over time. Indeed, the fact that $\|E\| = O(\epsilon)$ throughout training is key to both the proofs of odd and even $k^{\star}$, since it heuristically reduces our process to a Brownian component plus an $\epsilon$ signal component that can leverage the randomness in the Brownian component.[1]

### 4.3 Recovery of $\theta^{\star}$

Let $\tilde{O}(\cdot)$ hide non-$\epsilon$ terms. In the odd case, our estimator converges to the direction of $\bar{b} = \mathbb{E}_{z \sim \mu}[b(z)]$ with a magnitude of $\tilde{O}(\epsilon)$. We prove in Section G that for the tensor PCA setting, this recovers $\theta^{\star}$ with $n \gtrsim d^{\lceil k^{\star}/2 \rceil}$, and we prove in Section H that for the single-index model setting, it recovers $\theta^{\star}$ with $n \gtrsim d^{\lceil k^{\star}/2 \rceil}$ as well. Moreover, we prove that when $n \gtrsim d^{k^{\star}/2}$, we obtain nontrivial correlation with $\theta^{\star}$, from which we can then run online SGD to get a total sample complexity of $d^{k^{\star}/2}$. For the even case, full proofs are included in Section G and Section G as well; we also leverage the uniform bound on $\sup \|E\|$ to prove convergence of our estimator $\hat{M}$ to approximately $\frac{I}{d}$ plus $\tilde{O}(\epsilon)$ spike in $\theta^{\star}\theta^{\star\top}$. From here, we can perform PCA or a similar algorithm to recover $\theta^{\star}$.

## 5 Discussion

### 5.1 Experiments

We sanity check our findings experimentally via different choices of link functions which correspond to different $k^{\star}$. For $k^{\star} = 3, 4, 5$, we let $\sigma(t) = h_{k^{\star}}(t)$, as defined in Definition 1. Specifically, we run the minibatch update defined in Section 5.2 with batch size 1. Our findings are included in Figure 1 and Figure 2 for the odd and even cases, respectively. For $k^{\star} = 3, 5$, our first-order estimator indeed recovers $\theta^{\star}$, even though the iterates stay near the equator throughout training. For $k^{\star} = 4$, this same estimator does not recover $\theta^{\star}$, but the second-order estimator's top eigendirection does, with the iterates once again staying near the equator. Our experiments are run with different learning rates, and we observe that smaller learning rates behave more and more like gradient flow, whereas larger ones behave more like Brownian motion and stay near the equator, as we would predict with Langevin dynamics. However, there are some more nuances to this, as we describe in the next section.

---

[1] As an aside, our technique is one way to prove convergence to the stationary Gibbs distribution $\mu_\epsilon \propto \exp(-2\epsilon L_n)$, and we believe this can be a useful way to approach our minibatch conjecture in Section 5.2.

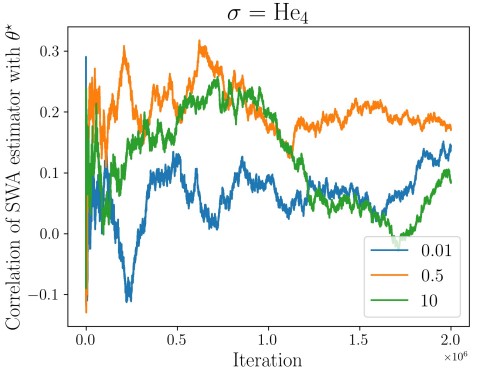
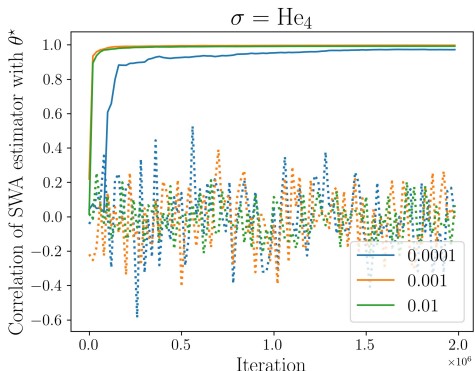

(a) For various learning rate choices, we track the time average (e.g. the first order estimator) as a function of iteration, which can be seen to not have any meaningful correlation with $\theta^\star$. This is due to the $\sigma'$ being an odd function, causing the first order estimator to vanish.

(b) The solid curves correspond to the correlation of $\theta^\star$ with the top eigenvector of the time average of $\theta\theta^\top$, and the dotted lines are for the correlation between the actual iterate $\theta$ and $\theta^\star$. Indeed, the actual iterate itself remains near the equator over all time.

Figure 2: Simulations for $k^\star = 4$, run with $d = 100$ with $n = 10d^2$ samples.

## 5.2 EXTENSION TO MINI-BATCH SGD

Our experimental results suggest that pure mini-batch SGD should have theoretically guarantees too. Consider mini-batch SGD with learning rate $\eta$ and batch size 1:

$$\theta_{t+1} = \frac{\theta_t - \eta g_t}{\|\theta_t - \eta g_t\|}, \quad g_t := \nabla_\theta L(\theta_t; x_{i_t}, y_{i_t}), \quad i_t \sim \mathcal{U}([n])$$

$g_t$ is approximately a standard Gaussian, since $\nabla L(\theta; x, y) = -y\sigma'(\theta \cdot x)x$ and $\theta \cdot x$ is $O(1)$ for the most part, and hence $\|g_t\| \approx O(\sqrt{d})$. For $\eta \ll d^{-1/2}$, we have the following approximation:

$$\theta_{t+1} = \frac{\theta_t - \eta g_t}{\|\theta_t - \eta g_t\|} = \frac{\theta_t - \eta g_t}{\sqrt{1 + \eta^2\|g_t\|^2}} \approx (\theta_t - \eta g_t)(1 - \frac{1}{2}\eta^2(d - 1))$$

Let $z_t := g_t + b(\theta_t)$ be the mini-batch noise[2]. Because we are in a noise-dominated regime, $z_t$ is approximately isotropic so if we approximate this process by an SDE, we would heuristically get:

$$\theta_{t+1} \approx \theta_t - \eta g_t - \frac{1}{2}\eta^2(d - 1)\theta_t$$

$$= \theta_t - \sqrt{\eta} \cdot \sqrt{\eta}z_t - \eta \cdot \frac{1}{2}\eta(d - 1)\theta + \eta b(\theta_t)$$

$$\implies d\theta \approx \left(-\frac{d-1}{2}\eta\theta + b(\theta)\right)dt + \sqrt{\eta}P_\theta^\perp dW_t$$

$$\implies d\theta \approx \left(-\frac{d-1}{2}\theta + \frac{1}{\eta}b(\theta)\right)dt + P_\theta^\perp dW_t$$

which roughly recovers our Langevin setting with $\epsilon := \frac{1}{\eta}$. *We therefore conjecture that there exists a learning rate regime for which this SGD argument holds even without the noise boosting that is present in Langevin dynamics.* The main technical challenge in extending our results in this direction is not just controlling the discretization error, but also the dependencies that arise between the noise covariance and the smoothing estimator. In particular, the stationary distribution for the pure-noise process will no longer be isotropic over the sphere and will have a data-dependent stationary distribution, which introduces additional complications. However, extending our results and techniques to the minibatch SGD setting is a promising direction for future work.

---

[2]By choosing batch size $B = 1$, we maximize the scale of the noise without explicit noise boosting.

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

## A  PRELIMINARIES

**Definition 7.** *Let $\iota = C_\iota \log(d)$ for a sufficiently large constant $C_\iota$. We define high probability events to be events that happen with probability at least $1 - \mathrm{poly}(d)e^{-\iota}$ where $\mathrm{poly}(d)$ does not depend on $C_\iota$.*

Note that high probability events are closed under polynomial number of union bounds.

**Lemma 4.** *The Itô stochastic differential equations for $\beta$ and $\theta$ remain on $S^{d-1}$ for all time.*

*Proof.* This follows by Itô's lemma on $f(X) = \frac{1}{2}\|X\|^2$. More concretely,

$$d\left(\frac{1}{2}\|\theta\|^2\right) = \left(-\frac{d-1}{2}(\theta \cdot \theta) + P_\theta^\perp \cdot \epsilon b(\theta)\theta + \frac{1}{2}\operatorname{tr} P_\theta^\perp\right)dt + \theta^\top P_\theta^\perp dW_t = 0$$

The derivation for $\beta$ proceeds similarly. $\square$

We proceed by applying Lemma 24 that gives high probability control of $E$ over all time.

**Lemma 5** (High probability uniform bound of $\sup\|E\|$). *With probability at least $1 - dTe^{-d}$, there exists an absolute constant $C'$ such that:*

$$\sup_{t \leq T}\|E(t)\| \leq C'\left[\frac{\epsilon \sup\|b\|}{d}\right]$$

*Proof.* Recall the SDE for $E(t)$:

$$dE = \left(-\frac{d-1}{2}E + \epsilon b(\theta)\right)dt + \left(P_\theta^\perp - P_\beta^\perp\right)dW_t$$

By Lemma 24, we can apply the result with $C = \frac{d-1}{2} \asymp d$, $G \asymp \epsilon \sup\|b\|$, and $B = 2$. $\square$

## B  ERGODIC CONCENTRATION

**Lemma 6** (Lemma 1, restated). *Let $f : \mathbb{R}^d \to \mathbb{R}^m$ such that $f \in L^2(\mu)$, where $\mu$ is the stationary uniform measure over the sphere for the Brownian motion, and $\int_{S^{d-1}} f d\mu = 0$. Then, we have:*

$$\frac{1}{T}\int_0^T f(\beta_t)dt = \frac{\phi(\beta_0) - \phi(\beta_T)}{T} + \frac{M_T}{T}$$

*where*

$$\phi(\beta) = \int_0^\infty P_t f(\beta)dt$$

*and $M_T := \int_0^T \nabla\phi(\beta_t)^\top P_{\beta_t}^\perp dW_t$ is a martingale.*

*Proof.* To begin, observe that $\phi$ satisfies $-\mathcal{L}\phi = f$. To see why, note that:

$$\mathcal{L}\phi(x) = \int_0^\infty \mathcal{L}(P_t f)(x)dt = [(P_t f)(x)]_0^\infty = -f(x)$$

where in the second equality we used Kolmogorov's backward equation:

$$\frac{d}{dt}P_t f = P_t \mathcal{L}f = \mathcal{L}P_t f, \quad P_0 f = f$$

Applying Itô's to $\phi(\beta_t)$, we obtain:

$$\begin{aligned}
d\phi(\beta) &= \nabla\phi(\beta) \cdot d\beta + \mathcal{L}\phi(\beta)dt \\
&= \nabla\phi(\beta)^\top P_\beta^\perp d\beta + \mathcal{L}\phi(\beta)dt \\
&= \nabla\phi(\beta)^\top P_\beta^\perp dW_t + \mathcal{L}\phi(\beta)dt
\end{aligned}$$

where the second line follows from that fact that $\beta^\top(d\beta) = 0$ (i.e. Brownian motion stays on the sphere). Therefore, it holds that by integrating from $0$ to $T$,

$$\phi(\beta_T) - \phi(\beta_0) = \int_0^T \nabla\phi(\beta_t)^\top P_{\beta_t}^\perp dW_t + \int_0^T \mathcal{L}\phi(\beta_t)dt$$

$$= M_T - \int_0^T f(\beta_t)dt$$

Rearranging gives the desired result. $\qquad\square$

**Lemma 7.** *In the setting of Lemma 6 with $f : \mathbb{R}^d \to \mathbb{R}^m$, the following holds:*

$$\left\| \frac{\phi(\beta_0) - \phi(\beta_T)}{T} \right\| \leq \frac{2 \sup \|\nabla f\|_2}{(d-2)T}$$

*Proof.* We recall that $\|\nabla f(\beta)\|_2$ can be interpreted as the Lipschitz constant of $f$ with respect to the Euclidean norm. First, note that two points on $S^{d-1}$ can differ by at most 2 in Euclidean norm. Therefore, we have:

$$\|\phi(\beta_0) - \phi(\beta_T)\| \leq 2 \sup \|\nabla\phi\|_2$$

We can then bound the supremum as follows:

$$\sup \|\nabla\phi(\beta)\|_2 = \sup \left\| \int_0^\infty \nabla P_t f(\beta)dt \right\|_2$$

$$\leq \int_0^\infty \sup \|\nabla P_t f(\beta)\|_2 dt$$

$$\leq \int_0^\infty e^{-(d-2)t} \sup \|\nabla f(\beta)\|_2 dt$$

$$= \frac{\sup \|\nabla f(\beta)\|_2}{d-2}$$

where the second to last inequality follows from the Ricci curvature of $S^{d-1}$ being $d-2$ and the gradient bound of Theorem 3.2.3 in Bakry et al. (2016), and the first result follows upon division by $T$. $\qquad\square$

**Lemma 8.** *In the setting of Lemma 6 with $f : \mathbb{R}^d \to \mathbb{R}^m$, the following holds with probability $1 - e^{-m}$:*

$$\left\| \frac{M_T}{T} \right\| \lesssim \sqrt{\frac{m \sup \|\nabla f(\beta)\|_2^2}{T(d-2)^2}}$$

*Proof.* Recall that $M_T := \int_0^T \nabla\phi(\beta_t)^\top P_{\beta_t}^\perp dW_t$. We consider the predictable quadratic variation matrix $\langle M_t \rangle = \int_0^T \nabla\phi(\beta_t)^\top P_{\beta_t}^\perp \left( \nabla\phi(\beta_t)^\top P_\beta^\perp \right)^\top dt$. Then, we have that:

$$\|\nabla\phi(\beta)^\top P_\beta^\perp\|_2 \leq \|\nabla\phi(\beta)\|_2 \leq \frac{\sup \|\nabla f(\beta)\|_2}{d-2}$$

Since we have operator norm control here (rather than Frobenius), applying Lemma 20 yields that with probability $1 - \delta$,

$$\|M_T\| \lesssim \frac{\sup \|\nabla f(\beta)\|_2}{d-2} \sqrt{T(m + \log(1/\delta))}$$

from which the desired result follows upon division by $T$. $\qquad\square$

**Corollary 2.** *In the setting of Lemma 6 with $f : \mathbb{R}^d \to \mathbb{R}^m$ it holds with probability $1 - e^{-m}$ that:*

$$\left\| \frac{1}{T} \int_0^T f(\beta_t)dt \right\| \lesssim \frac{\sup \|\nabla f(\beta)\|_2}{Td} + \sqrt{\frac{m \sup \|\nabla f(\beta)\|_2^2}{Td^2}}$$

## C    PROOF OF THE ODD $k^\star$ CASE

We now show that after sufficiently long running time, the time average of $\theta$ roughly approximates the time average of the Brownian motion, which in expectation over the stationary measure $\mu$ should converge to the partial trace estimator for $k^\star$ odd (i.e. $\mathbb{E}_{z\sim\mu}[b(z)]$).

**Proposition 1** (Decomposition of $E$). *At time $t \geq 0$, it holds that:*

$$E(t) = \int_0^t e^{-\frac{d-1}{2}(t-s)}\epsilon b(\theta_s)ds + \int_0^t e^{-\frac{d-1}{2}(t-s)}(P_\theta^\perp - P_\beta^\perp)dW_s$$

*Proof.* Recall the SDE's for the coupled processes $\theta$ and $\beta$.

$$d\theta = \left(-\frac{d-1}{2}\theta + \epsilon b(\theta)\right)dt + P_\theta^\perp dW_t$$

$$d\beta = -\frac{d-1}{2}\beta dt + P_\beta^\perp dW_t$$

This implies that:

$$dE = \left(-\frac{d-1}{2}E + \epsilon b(\theta)\right)dt + \left(P_\theta^\perp - P_\beta^\perp\right)dW_t$$

Integrating this gives the desired expression. □

We now give the ergodic concentration results for the relevant functions.

**Lemma 9** (Ergodic concentration of $b$). *Suppose $T \gtrsim d^{-1}$. With probability at least $1 - e^{-d}$, we have:*

$$\left\|\frac{1}{T}\int_0^T b(\beta_s)ds - \bar{b}\right\| \lesssim \frac{\sup\|\nabla b\|_2}{\sqrt{Td}} \lesssim \frac{1}{\sqrt{Td}} \tag{2}$$

*Proof.* This follows directly from Corollary 2, setting $f(\beta) = b(\beta) - \bar{b}$, and using the fact that $b$ is $O(1)$-Lipschitz. □

**Lemma 10** (Ergodic concentration of $\beta$). *Suppose $T \gtrsim d^{-1}$. With probability at least $1 - e^{-d}$, it holds that:*

$$\left\|\frac{1}{T}\int_0^T \beta_s ds\right\| \lesssim \frac{1}{\sqrt{Td}}$$

*Proof.* This follows directly from Corollary 2, setting $f(\beta) = \beta$. □

We now prove the main theorem.

**Theorem 4** (Theorem 2, restated). *Let $\epsilon = o\big(d^{-(k^\star-3)/2}\big)$ and $T \gtrsim d^{k^\star}/\epsilon^2$. Then for $\delta, \Delta > 0$, if $n \gtrsim d^{\lceil k^\star/2\rceil}/\Delta^2$, Algorithm 1 succeeds in recovering the ground truth $\theta^\star$ up to error $\Delta$ with probability at least $1 - e^{d^c}$.*

*Proof.* The time average of the $E$ up to time $T$ is the sum of the time averages of the two terms in Proposition 1. For the second term, which is the noise term, we have the following:

$$\begin{aligned}
M_T &:= \frac{1}{T}\int_0^T \int_0^t e^{-\frac{d-1}{2}(t-s)}(P_\theta^\perp - P_\beta^\perp)dW_s dt \\
&= \frac{1}{T}\int_0^T (P_\theta^\perp - P_\beta^\perp)\int_0^{T-s} e^{-\frac{d-1}{2}t}dt dW_s \\
&= \frac{1}{T}\int_0^T (P_\theta^\perp - P_\beta^\perp)\cdot\frac{2}{d-1}\left(1 - e^{-\frac{d-1}{2}(T-s)}\right)dW_s
\end{aligned}$$

Note that $\|P_\theta^\perp - P_\beta^\perp\|_F \lesssim \sup\|E\| \lesssim \frac{\epsilon}{d}$. Therefore, by Lemma 21, we have that with probability $1 - e^{-d}$,

$$\left\|\frac{M_T}{T}\right\| \lesssim \frac{\epsilon}{\sqrt{Td^3}}$$

For the first term in Proposition 1, we have

$$\frac{1}{T}\int_0^T \int_0^t e^{-\frac{d-1}{2}(t-s)}\epsilon b(\theta_s)dsdt = \frac{1}{T}\int_0^T \epsilon b(\theta_s)\int_0^{T-s} e^{-\frac{d-1}{2}t}dtds$$

$$= \frac{1}{T}\int_0^T \epsilon b(\theta_s)\cdot\frac{2}{d-1}\left(1 - e^{-\frac{d-1}{2}(T-s)}\right)ds$$

$$= \frac{1}{T}\int_0^T \epsilon b(\theta_s)\cdot\frac{2}{d-1}ds - \frac{1}{T}\int_0^T \epsilon b(\theta_s)\cdot\frac{2}{d-1}e^{-\frac{d-1}{2}(T-s)}ds$$

We analyze these two terms separately. For the second term, note that:

$$\left\|\frac{1}{T}\int_0^T \epsilon b(\theta_s)\cdot\frac{2}{d-1}e^{-\frac{d-1}{2}(T-s)}ds\right\| \lesssim \frac{\epsilon \sup\|b(\theta)\|}{Td}\int_0^T e^{-\frac{d-1}{2}(T-s)}ds \lesssim \frac{\epsilon \sup\|b(\theta)\|}{Td^2}$$

For the first term, we decompose it as follows to isolate the Brownian motion:

$$\frac{1}{T}\int_0^T \epsilon b(\theta_s)\cdot\frac{2}{d-1}ds = \frac{2}{T(d-1)}\int_0^T \epsilon b(\beta_s)ds + \frac{2}{T(d-1)}\int_0^T \epsilon(b(\theta_s) - b(\beta_s))ds$$

Once again, the second term can be bounded by the Lipschitz constant of $b$:

$$\left\|\frac{2}{T(d-1)}\int_0^T \epsilon(b(\theta_s) - b(\beta_s))ds\right\| \leq \frac{2\epsilon \sup\|\nabla b\|_2}{T(d-1)}\int_0^T \|\theta_s - \beta_s\|ds$$

$$\lesssim \frac{2\epsilon \sup\|\nabla b\|_2}{(d-1)}\left[\frac{\epsilon \sup\|b\|}{d}\right]$$

The remaining term is the main term $\frac{2\epsilon}{d-1}\frac{1}{T}\int_0^T b(\beta_s)ds$, which we proved concentration around the stationary average for in Lemma 9. Therefore, the time average of $E$ satisfies via triangle inequality:

$$\left\|\frac{1}{T}\int_0^T E_s ds - \frac{2\epsilon}{d-1}\bar{b}\right\|$$

$$\lesssim \left\|\frac{1}{T}\int_0^T \frac{2\epsilon}{d-1}(b(\beta) - \bar{b})ds\right\| + \frac{\epsilon}{\sqrt{Td^3}} + \frac{\epsilon \sup\|b\|}{Td^2} + \frac{2\epsilon^2 \sup\|\nabla b\|_2 \sup\|b\|}{d^2}$$

$$\lesssim \frac{2\epsilon}{d-1}\frac{\sup\|\nabla b\|_2}{\sqrt{Td}} + \frac{\epsilon}{\sqrt{Td^3}} + \frac{\epsilon \sup\|b\|}{Td^2} + \frac{2\epsilon^2 \sup\|\nabla b\|_2 \sup\|b\|}{d^2} \lesssim \frac{\epsilon}{\sqrt{Td^3}} + \frac{\epsilon^2}{d^2}$$

Combining our results with Lemma 10 using triangle inequality, we obtain with probability at least $1 - e^{-d}$:

$$\left\|\frac{1}{T}\int_0^T \theta_s ds - \frac{2\epsilon}{d-1}\bar{b}\right\| = \left\|\frac{1}{T}\int_0^T (\beta_s + E_s)ds - \frac{2\epsilon}{d-1}\bar{b}\right\|$$

$$\leq \left\|\frac{1}{T}\int_0^T \beta_s ds\right\| + \left\|\frac{1}{T}\int_0^T E_s ds - \frac{2\epsilon}{d-1}\bar{b}\right\|$$

$$\lesssim \frac{1}{\sqrt{Td}} + \frac{\epsilon}{\sqrt{Td^3}} + \frac{\epsilon^2}{d^2}$$

Let $u := \frac{2\epsilon}{d-1}\bar{b}$ and $v := \frac{1}{T}\int_0^T \theta_t dt$. Then, in our regime of $T$ and $\epsilon$, the total error is bounded as:

$$\|u - v\| \lesssim \frac{1}{\sqrt{Td}} + \frac{\epsilon}{\sqrt{Td^3}} + \frac{\epsilon^2}{d^2} \ll \frac{2\epsilon}{d-1} \cdot d^{-(k^\star - 1)/2}$$

By our lemma, we have that with probability $1 - e^{-d^c}$:

$$\left\|\bar{b} - \mathbb{E}_x[\bar{b}]\right\| \lesssim \Delta d^{-(k^\star - 1)/2}$$

We wish to analyze $\frac{v \cdot \theta^\star}{\|v\|}$, which we calculate via triangle inequality as:

$$
\begin{aligned}
\frac{v \cdot \theta^\star}{\|v\|} &\geq \frac{\frac{2\epsilon}{d-1}\mathbb{E}_x[\bar{b}] \cdot \theta^\star - \left\|\frac{2\epsilon}{d-1}\bar{b} - \frac{2\epsilon}{d-1}\mathbb{E}_x[\bar{b}]\right\| - \left\|v - \frac{2\epsilon}{d-1}\bar{b}\right\|}{\left\|\frac{2\epsilon}{d-1}\mathbb{E}_x[\bar{b}]\right\| + \left\|\frac{2\epsilon}{d-1}\bar{b} - \frac{2\epsilon}{d-1}\mathbb{E}_x[\bar{b}]\right\| + \left\|v - \frac{2\epsilon}{d-1}\bar{b}\right\|} \\
&\geq \frac{\frac{2\epsilon}{d-1}(1 - \Delta)}{\frac{2\epsilon}{d-1}(1 + \Delta)} \\
&\geq 1 - \Delta
\end{aligned}
$$

as desired. $\qquad\square$

## D  PROOF OF THE EVEN $k^\star$ CASE

**Lemma 11** (Ergodic concentration of $\beta\beta^\top$). *Suppose $T \gtrsim d^{-2}$. With probability at least $1 - e^{-d}$, it holds that:*

$$\left\|\frac{1}{T}\int_0^T \beta_s \beta_s^\top ds - \frac{I}{d}\right\|_F \lesssim \frac{1}{\sqrt{T}}$$

*Proof.* This follows directly from Corollary 2, setting $f(\beta) = \beta\beta^\top - \frac{I}{d}$, and flattening the matrix into a vector in $\mathbb{R}^{d^2}$. $\qquad\square$

**Lemma 12** (Ergodic concentration of $\beta b(\beta)^\top + b(\beta)\beta^\top$). *Suppose $T \gtrsim d^{-2}$. With probability at least $1 - e^{-d}$, we have that:*

$$\left\|\frac{1}{T}\int_0^T (\beta_s b(\beta_s)^\top + b(\beta_s)\beta_s^\top)ds - \mathbb{E}_{z \sim \mu}[zb(z)^\top + b(z)z^\top]\right\|_F \lesssim \frac{\sup \|\nabla(\beta b(\beta)^\top)\|_2 + 1}{\sqrt{T}} \lesssim \frac{1}{\sqrt{T}}$$

*Proof.* This follows directly from Corollary 2, setting $f(\beta) = \beta b(\beta)^\top + b(\beta)\beta^\top - \mathbb{E}_{z \sim \mu}[zb(z)^\top + b(z)z^\top]$, and flattening the matrix into a vector in $\mathbb{R}^{d^2}$. $\qquad\square$

**Lemma 13.** *With probability $1 - e^{-d^c}$, it holds that:*

$$\left\|\frac{1}{T}\int_0^T (E_s b(\theta_s)^\top + b(\theta_s)E_s^\top)ds - \frac{\epsilon}{d}\mathbb{E}_{z \sim \mu}[zb(z)^\top + b(z)z^\top]\right\|_F \lesssim \frac{\epsilon}{d\sqrt{T}} + \frac{\epsilon^2}{d^2}$$

*Proof.* Recall the SDE's for $E$ and $\beta$:

$$d\beta = -\frac{d-1}{2}\beta dt + P_\beta^\perp dW_t$$

$$dE = \left(-\frac{d-1}{2}E + \epsilon b(\theta)\right)dt + \left(P_\theta^\perp - P_\beta^\perp\right)dW_t$$

By Itô's lemma, we calculate the SDE for $E\beta^\top$ as:

$$d(E\beta^\top) = \left(-(d-1)E\beta^\top + \epsilon b(\theta)\beta^\top + (P_\theta^\perp - P_\beta^\perp)P_\beta^\perp\right)dt + (P_\theta^\perp - P_\beta^\perp)dW_t\beta^\top + EdW_t^\top P_\beta^\perp$$

The SDE of $\beta E^\top$ is just the transpose of the above, so we have:

$$d(E\beta^\top) = \left(-(d-1)\beta E^\top + \epsilon\beta b(\theta)^\top + P_\beta^\perp(P_\theta^\perp - P_\beta^\perp)\right)dt + \beta dW_t^\top(P_\theta^\perp - P_\beta^\perp) + P_\beta^\perp dW_t E^\top$$

Let $G := E\beta^\top + \beta E^\top$. Then the SDE for $G$ is:

$$
\begin{aligned}
d(G) &= \left(-(d-1)G + \epsilon(b(\theta)\beta^\top + \beta b(\theta)^\top) + \left[(P_\theta^\perp - P_\beta^\perp)P_\beta^\perp + P_\beta^\perp(P_\theta^\perp - P_\beta^\perp)\right]\right)dt \\
&\quad + (P_\theta^\perp - P_\beta^\perp)dW_t\beta^\top + EdW_t^\top P_\beta^\perp + \beta dW_t^\top(P_\theta^\perp - P_\beta^\perp) + P_\beta^\perp dW_t E^\top
\end{aligned}
$$

where the first line is the drift term, and the second line is the noise term. Moreover, we can further simplify the final term in the drift:

$$
\begin{aligned}
&(P_\theta^\perp - P_\beta^\perp)P_\beta^\perp + P_\beta^\perp(P_\theta^\perp - P_\beta^\perp) \\
&= (-\beta E^\top - EE^\top + (E^\top\beta)(\beta\beta^\top + E\beta^\top)) + (-E\beta^\top - EE^\top + (E^\top\beta)(\beta\beta^\top + \beta E^\top)) \\
&= -(\beta E^\top + E\beta^\top) + \Xi
\end{aligned}
$$

where $\Xi$ is the remainder term satisfying $\|\Xi\|_F \lesssim \|E\|^2 \lesssim \epsilon^2/d^2$. The last line follows from Lemma 14 for simplification. Our SDE for $G$ can therefore be rewritten as:

$$
\begin{aligned}
dG &= \left(-dG + \epsilon(b(\theta)\beta^\top + \beta b(\theta)^\top) + \Xi\right)dt \\
&\quad + (P_\theta^\perp - P_\beta^\perp)dW_t\beta^\top + EdW_t^\top P_\beta^\perp + \beta dW_t^\top(P_\theta^\perp - P_\beta^\perp) + P_\beta^\perp dW_t E^\top
\end{aligned}
$$

This implies that:

$$
\begin{aligned}
G(t) &= \int_0^t e^{-d(t-s)}\left(\epsilon(b(\theta_s)\beta_s^\top + \beta_s b(\theta_s)^\top) + \Xi_s\right)ds \\
&\quad + \int_0^t e^{-d(t-s)}\left[(P_\theta^\perp - P_\beta^\perp)dW_t\beta^\top + EdW_t^\top P_\beta^\perp + \beta dW_t^\top(P_\theta^\perp - P_\beta^\perp) + P_\beta^\perp dW_t E^\top\right]
\end{aligned}
$$

We first analyze the time average of the second term, which is the noise term. Intuitively, the time average of it should concentrate around 0 as time increases.

$$
\begin{aligned}
&\frac{1}{T}\int_0^T \int_0^t e^{-d(t-s)}\left[(P_\theta^\perp - P_\beta^\perp)dW_s\beta_s^\top + EdW_s^\top P_\beta^\perp + \beta dW_s^\top(P_\theta^\perp - P_\beta^\perp) + P_\beta^\perp dW_s E^\top\right]dt \\
&= \frac{1}{T}\int_0^T (P_\theta^\perp - P_\beta^\perp)\int_0^{T-s} e^{-dt}dt\,dW_s\beta_s^\top + \frac{1}{T}\int_0^T E\int_0^{T-s} e^{-dt}dt\,dW_s^\top P_\beta^\perp \\
&\quad + \frac{1}{T}\int_0^T \beta\int_0^{T-s} e^{-dt}dt\,dW_s(P_\theta^\perp - P_\beta^\perp) + \frac{1}{T}\int_0^T P_\beta^\perp\int_0^{T-s} e^{-dt}dt\,dW_s E^\top \\
&= \frac{1}{T}\int_0^T (P_\theta^\perp - P_\beta^\perp)\left(\frac{1}{d}(1 - e^{-d(T-s)})\right)dW_s\beta_s^\top + \frac{1}{T}\int_0^T E\left(\frac{1}{d}(1 - e^{-d(T-s)})\right)dW_s^\top P_\beta^\perp \\
&\quad + \frac{1}{T}\int_0^T \beta\left(\frac{1}{d}(1 - e^{-d(T-s)})\right)dW_s(P_\theta^\perp - P_\beta^\perp) + \frac{1}{T}\int_0^T P_\beta^\perp\left(\frac{1}{d}(1 - e^{-d(T-s)})\right)dW_s E^\top
\end{aligned}
$$

It now suffices to bound the Frobenius norm of the time average of the top two terms of the last expression (since the latter two terms are just transposes). We again observe that $\|P_\theta^\perp - P_\beta^\perp\|_F \lesssim \sup\|E\| \lesssim \frac{\epsilon}{d}$. For the first term, we have that:

$$
\begin{aligned}
&\mathbb{E}\left[\left\|\frac{1}{T}\int_0^T (P_\theta^\perp - P_\beta^\perp)\left(\frac{1}{d}(1 - e^{-d(T-s)})\right)dW_s\beta_s^\top\right\|_F^2\right] \\
&\lesssim \frac{1}{T^2}\int_0^T \mathbb{E}\left[\left(\frac{1}{d}(1 - e^{-d(T-s)})\right)^2 \|P_\theta^\perp - P_\beta^\perp\|_F^2\right]ds \\
&\lesssim \frac{1}{d^2 T}\sup_{t\leq T}\|E_t\|^2 \\
&\lesssim \frac{\epsilon^2}{d^4 T}
\end{aligned}
$$

where the second to last inequality follows from Lemma 15.

For the second term in the time average of the noise component, we have:

$$
\mathbb{E}\left[\left\|\frac{1}{T}\int_0^T E\left(\frac{1}{d}(1-e^{-d(T-s)})\right)dW_s^\top P_\beta^\perp\right\|_F^2\right]
$$

$$
\leq \frac{1}{T^2}\int_0^T \mathbb{E}\left[\left(\frac{1}{d}(1-e^{-d(T-s)})\right)^2\|E\|_F^2\right]
$$

$$
\lesssim \frac{\epsilon^2}{d^4 T}
$$

Combining all four noise terms together using Lemma 21 and triangle inequality, we have that with probability $1-e^{-d}$,

$$
\left\|\frac{1}{T}\int_0^T\int_0^t e^{-d(t-s)}\left[(P_\theta^\perp - P_\beta^\perp)dW_s\beta_s^\top + EdW_s^\top P_\beta^\perp + \beta dW_s^\top(P_\theta^\perp - P_\beta^\perp) + P_\beta^\perp dW_s E^\top\right]dt\right\|_F \lesssim \frac{\epsilon}{\sqrt{T d^3}}
$$

We now analyze the drift term of $G$. First, to isolate the Brownian motion, we once again do another decomposition:

$$
\int_0^t e^{-d(t-s)}\left(\epsilon(b(\theta_s)\beta_s^\top + \beta_s b(\theta_s)^\top) + \Xi_s\right)ds
$$

$$
= \int_0^t e^{-d(t-s)}\left(\epsilon((b(\beta_s)+v)\beta_s^\top + \beta_s(b(\beta_s)+v)^\top) + \Xi_s\right)ds
$$

$$
= \int_0^t e^{-d(t-s)}\epsilon(b(\beta_s)\beta_s^\top + \beta_s b(\beta_s)^\top)ds + \int_0^t e^{-d(t-s)}\left(\epsilon(v\beta_s^\top + \beta_s v^\top) + \Xi_s\right)ds
$$

where here we define $v := b(\theta) - b(\beta)$, which by Lipschitzness has norm bounded by $O(\|E\|) \lesssim \frac{\epsilon}{d}$. Hence, for all $t \leq T$, this second term satisfies:

$$
\left\|\int_0^t e^{-d(t-s)}\left(\epsilon(v\beta_s^\top + \beta_s v^\top) + \Xi_s\right)ds\right\|_F \leq \frac{1}{d}\sup_{s\leq t}\|\epsilon(v\beta_s^\top + \beta_s v^\top) + \Xi_s\|_F \lesssim \epsilon^2/d^2
$$

which means the time average over this component also has Frobenius norm $O(\epsilon^2/d^2)$. For the time average of the first term, we have the following:

$$
\frac{1}{T}\int_0^T\int_0^t e^{-d(t-s)}\epsilon(b(\beta_s)\beta_s^\top + \beta_s b(\beta_s)^\top)ds
$$

$$
= \frac{1}{T}\int_0^T\left(\frac{1}{d}(1-e^{-d(T-s)})\right)\epsilon(b(\beta_s)\beta_s^\top + \beta_s b(\beta_s)^\top)ds
$$

$$
= \frac{1}{T}\int_0^T\frac{1}{d}\epsilon(b(\beta_s)\beta_s^\top + \beta_s b(\beta_s)^\top)ds - \frac{1}{T}\int_0^T\frac{1}{d}e^{-d(T-s)}\epsilon(b(\beta_s)\beta_s^\top + \beta_s b(\beta_s)^\top)ds
$$

For the second term, we can bound this in Frobenius norm by:

$$
\left\|\frac{1}{T}\int_0^T\frac{1}{d}e^{-d(T-s)}\epsilon(b(\beta_s)\beta_s^\top + \beta_s b(\beta_s)^\top)ds\right\|_F \leq \frac{\epsilon}{Td}\sup\|b(\beta)\beta^\top + \beta b(\beta)^\top\|_F \lesssim \frac{\epsilon}{Td}
$$

Finally, for the first term, we have shown concentration to $\frac{\epsilon}{d}\mathbb{E}_{z\sim S^{d-1}}[b(z)z^\top + zb(z)^\top]$ in the previous lemma. Combining everything through triangle inequality, we have:

$$\left\| \frac{1}{T}\int_0^T G(s)ds - \frac{\epsilon}{d}\mathbb{E}_{z\sim\mu}[zb(z)^\top + b(z)z^\top] \right\|_F \lesssim \frac{\epsilon}{d}\left\| \frac{1}{T}\int_0^T \beta_s b(\beta_s) + b(\beta_s)\beta_s^\top ds - \mathbb{E}_{z\sim\mu}[zb(z)^\top + b(z)z^\top] \right\|_F$$

$$+ \frac{\epsilon}{Td} + \frac{\epsilon^2}{d^2} + \frac{\epsilon}{\sqrt{Td^3}}$$

$$\lesssim \frac{\epsilon}{d\sqrt{T}} + \frac{\epsilon}{Td} + \frac{\epsilon^2}{d^2} + \frac{\epsilon}{\sqrt{Td^3}}$$

$$\lesssim \frac{\epsilon}{d\sqrt{T}} + \frac{\epsilon^2}{d^2}$$

and the result follows. $\qquad\square$

**Theorem 5** (Theorem 3, restated). *Let $\epsilon = o(d^{-(k^\star-2)/2})$, and let $T \gtrsim d^{k^\star+2}/\epsilon^2$. Then, for $\Delta > 0$, if $n \gtrsim d^{k^\star/2}/\Delta^2$, the algorithm succeeds in recovering $\theta^\star$ up to error $\Delta$ with probability at least $1 - e^{-d^c}$.*

*Proof.* Recall that $\theta\theta^\top = \beta\beta^\top + E\beta^\top + \beta E^\top + EE^\top$. In the previous lemmas, we have analyzed each of these terms separately, and our goal is to prove ergodic concentration to $\frac{1}{d}I + \frac{\epsilon}{d}\mathbb{E}_{z\sim S^{d-1}}[zb(z)^\top + b(z)z^\top]$.

$$\left\| \frac{1}{T}\int_0^T \theta_s\theta_s^\top ds - \left(\frac{1}{d}I + \frac{\epsilon}{d}\mathbb{E}_{z\sim S^{d-1}}[zb(z)^\top + b(z)z^\top]\right) \right\|_F$$

$$\leq \left\| \frac{1}{T}\int_0^T \beta_s\beta_s^\top ds - \frac{I}{d} \right\|_F + \left\| \frac{1}{T}\int_0^T (E\beta^\top + \beta E^\top)ds - \frac{\epsilon}{d}\mathbb{E}_{z\sim S^{d-1}}[zb(z)^\top + b(z)z^\top]) \right\|_F + \left\| \frac{1}{T}\int_0^T EE^\top ds \right\|_F$$

$$\lesssim \frac{1}{\sqrt{T}} + \frac{\epsilon}{d\sqrt{T}} + \frac{\epsilon^2}{d^2} + \frac{\epsilon^2}{d^2}$$

$$\asymp \frac{1}{\sqrt{T}} + \frac{\epsilon}{d\sqrt{T}} + \frac{\epsilon^2}{d^2}$$

Consider the stationary average of $M_n := \frac{1}{d}I + \frac{\epsilon}{d}\mathbb{E}_{z\sim S^{d-1}}[zb(z)^\top + b(z)z^\top]$. By Lemma 30, with probability $1 - e^{-d}$, it holds that:

$$\left\| \mathbb{E}_{z\sim S^{d-1}}[zb(z)^\top + b(z)z^\top] - \mathbb{E}_{z\sim S^{d-1},x}[zb(z)^\top + b(z)z^\top] \right\|_2 \lesssim \sqrt{d^{-k^\star/2}/n}$$

Therefore, we obtain via triangle inequality that:

$$\left\| \frac{1}{T}\int_0^T \theta_s\theta_s^\top ds - \mathbb{E}_x[M_n] \right\|_2 \leq \left\| \frac{1}{T}\int_0^T \theta_s\theta_s^\top ds - M_n \right\|_2 + \|M_n - \mathbb{E}_x[M_n]\|_2$$

$$\lesssim \frac{1}{\sqrt{T}} + \frac{\epsilon}{d\sqrt{T}} + \frac{\epsilon^2}{d^2} + \frac{\epsilon}{d}\sqrt{d^{-k^\star/2}/n}$$

$$\lesssim \frac{\epsilon}{d}\sqrt{d^{-k^\star/2}/n}$$

where the last inequality follows from our regime of $\epsilon$ and $T$. We now note that the eigengap for $\mathbb{E}_x[M_n]$ is $\frac{\epsilon}{d}\Theta(d^{-k^\star/2})$. Then, when $n = \Theta(d^{k^\star/2}/\Delta^2)$, when applying Davis-Kahan, we see that the top eigenvector can be recovered up to accuracy:

$$\sin(u_1, \theta^\star) \lesssim \frac{\frac{\epsilon}{d}\sqrt{d^{-k^\star/2}/n}}{\frac{\epsilon}{d}\Theta(d^{-k^\star/2})} \lesssim \Delta$$

where $u_1$ denotes the top eigenvector of our time averaged matrix. $\qquad\square$

# E USEFUL LEMMAS

**Lemma 14.** *Let $\beta, \beta' \in S^{d-1}$, and let $E = \beta - \beta'$. Then, we have that*

$$E^\top \beta' = -\frac{1}{2}\|E\|^2$$

*Proof.*

$$\|\beta' + E\|^2 = \|\beta\|^2 \implies 2E^\top \beta' + \|E\|^2 = 0$$

since $\|\beta\| = \|\beta'\| = 1$. Rearranging gives the desired result. □

**Lemma 15.** *Let $\beta, \beta' \in S^{d-1}$. Then, we have that*

$$\mathrm{tr}\left((P_\beta^\perp - P_{\beta'}^\perp)(P_\beta^\perp - P_{\beta'}^\perp)^\top\right) = 2\|E\|^2 - \frac{1}{2}\|E\|^4$$

*where $E = \beta - \beta'$.*

*Proof.*

$$\mathrm{tr}\left((P_\beta^\perp - P_{\beta'}^\perp)(P_\beta^\perp - P_{\beta'}^\perp)^\top\right) = \mathrm{tr}\left(P_\beta^\perp(\beta'\beta'^\top) + P_{\beta'}(\beta\beta^\top)\right)$$

Note that

$$\begin{aligned}
P_{\beta'}^\perp(\beta\beta^\top) &= P_{\beta'}^\perp(\beta'\beta'^\top + \beta'E^\top + E\beta'^\top + EE^\top) \\
&= P_{\beta'}^\perp(E\beta'^\top + EE^\top) \\
&= E\beta'^\top + EE^\top - \beta'\beta'^\top E\beta'^\top - \beta'\beta'^\top EE^\top
\end{aligned}$$

and similarly

$$P_\beta^\perp(\beta'\beta'^\top) = -E\beta^\top + EE^\top + \beta\beta^\top E\beta^\top - \beta\beta^\top EE^\top$$

Summing these, we get the trace to be

$$2\|E\|^2 - 1/2\|E\|^4$$

□

**Lemma 16.** *Let $z \sim S^{d-1}$. Then, for integers $k \geq 0$, it holds that:*

$$\mathbb{E}_z\left[z_1^{2k}\right] = \frac{(2k-1)!!}{\prod_{j=0}^{k-1}(d+2j)} = \Theta(d^{-k})$$

**Lemma 17.** *Suppose $f(x)$ has information exponent $k^\star \geq 1$. Then, the information exponent of $g(x) := xf(x)$ has information exponent $k^\star - 1$.*

**Lemma 18.** *Let $g = \sum_k c_k h_k$ where $h_k$ is the $k$-th normalized Hermite polynomial and let $\ell$ be the index of the first nonzero even coefficient. Then,*

$$\mathbb{E}\left[(\mathbb{E}_z g(z \cdot x))^2\right] \lesssim \mathbb{E}_{x \sim N(0,1)}[g(x)^2]d^{-\ell/2}.$$

*Proof.* Note that we can rearrange this as:

$$\mathbb{E}_{z,z',x}[g(z \cdot x)g(z' \cdot x)] = \sum_k c_k^2 \mathbb{E}_{z,z'}[(z \cdot z')^k] = \sum_k c_{2k}^2 \mathbb{E}_{z,z'}[(z \cdot z')^{2k}].$$

We can now upper bound this by:

$$\mathbb{E}_{x \sim N(0,1)}[g(x)^2]\mathbb{E}_{z,z'}\left[\sum_{k \geq \ell/2}(z \cdot z')^{2k}\right] = \mathbb{E}_{x \sim N(0,1)}[g(x)^2]\mathbb{E}\left[\frac{(z \cdot z')^\ell}{1 - (z \cdot z')^2}\right].$$

The result now follows from (Damian et al., 2023, Lemma 26). □

## F    MISCELLANEOUS CONCENTRATION INEQUALITIES

**Lemma 19** (Concentration of norm). *Let $Z \sim \mathcal{N}(0, I_d)$. Then, it holds that:*

$$\Pr[\|Z\| - \mathbb{E}[\|Z\|] \geq s] \leq \exp(-s^2/2)$$

**Lemma 20.** *Suppose $M_T = \int_0^T A_t dW_t$ is a vector martingale in $\mathbb{R}^d$, with $\|A_t\|_2 \leq \alpha$ for all $t$. Then, it holds that:*

$$\mathbb{P}\left[\|M_T\| \geq \alpha\sqrt{T}\left(\sqrt{d} + \sqrt{2\log\frac{1}{\delta}}\right)\right] \leq \delta$$

**Lemma 21.** *In the setting of Lemma 20, suppose we instead have Frobenius norm control (e.g. $\|A_t\|_F \leq \alpha$ for all $t$). Then, it holds that:*

$$\mathbb{P}\left[\|M_T\| \geq \alpha\sqrt{T}\left(1 + \sqrt{2\log\frac{1}{\delta}}\right)\right] \leq \delta$$

**Lemma 22.** *Let $X : \mathbb{R} \to \mathbb{R}$ satisfy $X(0) = 0$ and*

$$dX = -AX\,dt + \sigma(X)dW_t.$$

*If $\sigma(X) \leq \sigma$ for all $X$, then for all $0 \leq s \leq t$, it holds that $X(t) - X(s)$ is $\frac{\sigma^2}{C}\left(1 - e^{-2C(t-s)}\right)$-subgaussian.*

*Proof.* Let $Y(t) := e^{At}X_t$. Then,

$$dY(t) = e^{At}\sigma(X(t))dW_t$$

Thus, $Y(t)$ is a martingale. Furthermore, the quadratic variation of $Y$ satisfies

$$\langle Y \rangle_t = \int_0^t e^{2At}\sigma(X(t))^2 dt \leq \sigma^2\int_0^t e^{2At}dt = \sigma^2 \cdot \frac{e^{2At} - 1}{2A} < \infty$$

Therefore, Novikov's condition tells us that

$$\mathcal{E}(\lambda Y)_t := \exp\left(\lambda Y(t) - \frac{\lambda^2}{2}\langle Y \rangle_t\right)$$

is a martingale. Hence,

$$\mathcal{E}(\lambda Y)_s = \mathbb{E}[\mathcal{E}(\lambda Y)_t | \mathcal{F}_s] = \mathbb{E}\left[\exp\left(\lambda Y(t) - \frac{\lambda^2}{2}\langle Y \rangle_t\right)|\mathcal{F}_s\right]$$

Rearranging the above inequality gives us

$$\mathbb{E}[\exp(\lambda Y(t))|\mathcal{F}_s]$$
$$\leq \mathbb{E}\left[\exp\left(\lambda Y(s) + \frac{\lambda^2\sigma^2}{2}\frac{e^{2At} - e^{2As}}{2A}\right)|\mathcal{F}_s\right]$$

Now, converting back to $X$ and replacing $\lambda \leftarrow \lambda e^{-At}$, we obtain

$$\mathbb{E}[\exp(\lambda(X(t) - X(s)))|\mathcal{F}_s]$$
$$\leq \mathbb{E}\left[\exp\left(\lambda X(s)(e^{-A(t-s)} - 1) + \frac{\lambda^2\sigma^2}{2}\frac{1 - e^{-2A(t-s)}}{2A}\right)|\mathcal{F}_s\right]$$

Applying this for $(s, 0)$ instead of $(t, s)$ gives us

$$\mathbb{E}[\exp(\lambda X(s))] \leq \exp\left(\frac{\lambda^2\sigma^2}{2}\frac{1 - e^{-2As}}{2A}\right) \leq \exp\left(\frac{\lambda^2\sigma^2}{4A}\right)$$

Plugging this in the previous equation upon taking expectation over $\mathcal{F}_s$, we obtain

$$\mathbb{E}[\exp(\lambda(X(t) - X(s)))] \leq \exp\left(\frac{\lambda^2\sigma^2(e^{-A(t-s)} - 1)^2}{4A} + \frac{\lambda^2\sigma^2(1 - e^{-2A(t-s)})}{4A}\right)$$

$$\leq \exp\left(\frac{\lambda^2\sigma^2}{2A}(1 - e^{-2A(t-s)})\right)$$

where we substituted and used the fact that

$$(e^{-A(t-s)} - 1)^2 \leq 1 - e^{-2A(t-s)}$$

$\square$

**Lemma 23** (Chaining tail inequality (van Handel, 2016)). *Let $\{X_t\}_{t\in T}$ be a separable subgaussian process on the metric space $(T, d)$. Then for all $t_0 \in T$ and $x \geq 0$,*

$$\Pr\left[\sup_{t\in T}\{X_t - X_{t_0}\} \geq C \int_0^\infty \sqrt{\log N(T, d, \epsilon)}d\epsilon + x\right] \leq Ce^{-\frac{x^2}{C\,\text{diam}(T)^2}}$$

*where $C < \infty$ is a universal constant, and $N(T, d, \epsilon)$ denotes the covering number of an $\epsilon$-net for $(T, d)$.*

**Corollary 3.** *In the setting of Lemma 22, there exists an absolute constant $C < \infty$ such that for any $\delta > 0$,*

$$\Pr\left[\sup_{t\leq T} |X_t| \geq C \times \frac{\sigma}{\sqrt{A}}\sqrt{\log\frac{1 + AT}{\delta}}\right] \leq \delta$$

*Proof.* Define

$$d(s, t) := \sqrt{\frac{\sigma^2}{A}(1 - e^{-2A(t-s)})}$$

Then, $X_t - X_s$ is $d(s,t)$-subgaussian from the Lemma 22. When we invert this distance, we obtain

$$N([0, T], d, \epsilon) \lesssim \frac{2AT}{-\log\left(1 - \frac{A\epsilon^2}{\sigma^2}\right)}$$

Note that for $\epsilon < \sigma/\sqrt{A}$, this can be upper bounded by $1 + \frac{2T\sigma^2}{\epsilon^2}$ and the diameter is upper bounded by $\sigma/\sqrt{A}$. Applying the chaining tail inequality in Lemma 23, we have:

$$\Pr\left[\sup_{t\leq T} \|X_t\| \geq C \times \frac{\sigma}{\sqrt{A}}\sqrt{\log(1 + AT)} + x\right] \leq e^{-\frac{x^2 A}{C'\sigma^2}}$$

where we used the fact that:

$$\int_0^\infty \sqrt{\log N([0, T], d, \epsilon)}d\epsilon \lesssim \frac{R}{\sqrt{A}}\sqrt{\log(1 + AT)}$$

Rearranging gives the desired result. $\square$

**Lemma 24.** *Let $X(0) = 0$ and suppose $X$ satisfies the following SDE.*

$$dX = [-AX + b(X)]dt + \Sigma^{1/2}(X)dW_t$$

*and that uniformly for all $X$,*

$$\|b(X)\| \leq G, \quad \text{tr}\,\Sigma(X) \leq B\|X\|^2$$

*Then, there exists an absolute constant $C > 0$ such that for any $\delta, T > 0$, if $L := 1 \vee \log\frac{1+AT}{\delta}$ and $A \geq CBL$, then with probability at least $1 - \delta$:*

$$\sup_{t\leq T} \|X(t)\| \leq \frac{CG}{A}.$$

*Proof.* We begin by decomposing $X(t) = X_1(t) + X_2(t)$ where $X_1, X_2$ follow:

$$dX_1 = [-AX_1 + b(X)]dt, \quad dX_2 = -AX_2 dt + \Sigma^{1/2}(X)dW_t$$

and $X_1(0) = X_2(0) = 0$. Define $R := \frac{G}{A}$. Observe that for all $t$,

$$X_1(t) = \int_0^t e^{-A(t-s)} b(X(s))ds \implies \|X_1(t)\| \le G \int_0^t e^{-A(t-s)}ds \le \frac{G}{A} = R.$$

For $X_2$, note that:

$$d\|X_2\|^2 = [-2A\|X_2\|^2 + \operatorname{tr}\Sigma(X)]dt + X_2^\top \Sigma^{1/2}(X)dW_t$$

We now decompose $\|X_2\|^2 = Y_1 + Y_2$ so that:

$$dY_1 = [-2AY_1 + \operatorname{tr}\Sigma(X)]dt, \quad dY_2 = -2AY_2 dt + X_2^\top \Sigma^{1/2}(X)dW_t.$$

Define the stopping time $\tau := \inf\{t \ge 0 \ : \ \|X_2(t)\| \ge R\}$. Then

$$\operatorname{tr}\Sigma(X(t \wedge \tau)) \le B\|X(t \wedge \tau)\|^2 \le 2B\left[\frac{G^2}{A^2} + R^2\right] = 4BR^2.$$

Therefore $Y_1(t \wedge \tau) \le 2BR^2/A$. Next, the noise term in the SDE for $Y_2$ can be bounded by:

$$X_2(t \wedge \tau)^T \Sigma(X(t \wedge \tau)) X_2(t \wedge \tau) \le \|X_2(t \wedge \tau)\|^2 \operatorname{tr}\Sigma(X(t \wedge \tau)) \le 4BR^4.$$

Now, let $C$ be a sufficiently large constant. Substituting into Corollary 3, we have that with probability at least $1 - \delta$,

$$\sup_{t \le T} \|Y_2(t \wedge \tau)\| \le C\sqrt{\frac{BR^4}{A} \log\left(\frac{2(1+AT)}{\delta}\right)}.$$

Under this event, we have that

$$\sup_{t \le T} \|X_2(t \wedge \tau)\|^2 \le CR^2\left[\frac{B}{A} + \sqrt{\frac{B}{A} \log\left(\frac{2(1+AT)}{\delta}\right)}\right].$$

Now since $A \ge C'B(1 \vee \log(1 + AT))$ where $C'$ is a sufficiently large constant then the right hand side is strictly less than $R$, which implies that with probability at least $1 - \delta$, $\tau < T$ and $\sup_{t \le T} \|X(t)\| \lesssim R$. $\qquad\square$

We now give the following standard definition of the Orlicz norm, which will be used extensively for our concentration results.

**Definition 8.** *For $\alpha > 0$, define the function $\psi_\alpha(t) = \exp(t^\alpha) - 1$. Then, for a random variable $X$, we define the $\psi_\alpha$ Orlicz norm of $X$ to be:*

$$\|X\|_{\psi_\alpha} = \inf\{\lambda > 0 : \mathbb{E}\psi_\alpha(|X|/\lambda) \le 1\}$$

*In particular, a mean zero random variable $X$ is $\|X\|_{\psi_1}$-subexponential, and is $\|X\|_{\psi_2}$-subgaussian.*

We now give the following lemma, which is adapted from Theorem 4 in (Adamczak, 2008) for our setting.

**Lemma 25** (Adapted from Theorem 4, (Adamczak, 2008))**.** *Suppose $X_1, \dots, X_n$ are i.i.d. random variables in a measureable space $(\mathcal{S}, \mathcal{B})$, and let $\mathcal{F}$ be a countable class of measureable function $f : \mathcal{S} \to \mathbb{R}$. Assume for every $f \in \mathcal{F}$, it holds that $\mathbb{E}f(X_i) = 0$ and that $\|\sup_{f \in \mathcal{F}} |f(X_i)|\|_{\psi_1} < \infty$. Define $Z := \sup_{f \in \mathcal{F}} |\sum_{i=1}^n f(X_i)|$ and $\sigma^2 := \sup_{f \in \mathcal{F}} \sum_{i=1}^n \mathbb{E}f(X_i)^2$. Then, with probability at least $1 - \delta$, it holds that:*

$$Z \lesssim \mathbb{E}Z + \sqrt{\sigma^2 \log\frac{1}{\delta}} + \|\max_i \sup_{f \in \mathcal{F}} |f(X_i)|\|_{\psi_1} \log\frac{1}{\delta}$$

**Lemma 26.** *Suppose $X_1, \ldots, X_n$ are i.i.d. mean-zero random variables on $\mathbb{R}^d$ such that for any $v \in S^{d-1}$, it holds that $\mathbb{E}[(X_i \cdot v)^2] \leq \sigma^2$ and $X_i \cdot v$ is $R$-subgaussian. Then, it holds with probability $1 - \delta$ that*

$$\left\| \frac{1}{n} \sum_i X_i \right\| \lesssim \sqrt{\frac{\sigma^2(d + \log(1/\delta))}{n}} + \frac{R \log(1/\delta)\sqrt{d + \log n}}{n}$$

*Proof.* Consider a $1/4$-net $\mathcal{N}_{1/4}$ of $S^{d-1}$. Define $Z = \sup_{v \in \mathcal{N}_{1/4}} \left| \frac{1}{n} \sum_i X_i \cdot v \right|$. Then, by Lemma 25, it holds that with probability at least $1 - \delta$:

$$Z \lesssim \mathbb{E}Z + \sqrt{\frac{\sigma^2 \log \frac{1}{\delta}}{n}} + \frac{\| \max_i \sup_{v \in \mathcal{N}_{1/4}} |X_i \cdot v| \|_{\psi_1} \log \frac{1}{\delta}}{n}$$

By union bound over $n \exp(d)$ terms with standard subgaussian tails, we have that:

$$\| \max_i \sup_{v \in \mathcal{N}_{1/4}} |X_i \cdot v| \|_{\psi_2} \lesssim R\sqrt{d + \log n}$$

Since the $\psi_1$ norm is upper bounded by the $\psi_2$ norm, we have that the above is an upper bound of the $\psi_1$ norm as well. For $\mathbb{E}Z$, we have that:

$$\mathbb{E}Z \leq \sqrt{\mathbb{E}[Z^2]} \leq \sqrt{\mathbb{E}\left\| \frac{1}{n} \sum_i X_i \right\|^2} = \sqrt{\operatorname{tr}\left( \operatorname{Cov}\left( \frac{1}{n} \sum_i X_i \right) \right)} \lesssim \sqrt{\frac{\sigma^2 d}{n}}$$

where in the second inequality we used the fact that $\mathcal{N}_{1/4} \subseteq S^{d-1}$. Combining everything with the covering argument, we have that with probability at least $1 - \delta$,

$$\left\| \frac{1}{n} \sum_i X_i \right\| \lesssim \sqrt{\frac{\sigma^2(d + \log \frac{1}{\delta})}{n}} + \frac{R\sqrt{d + \log n} \log \frac{1}{\delta}}{n}$$

as desired. $\qquad\square$

## G  TENSOR PCA

Let $T = (\theta^\star)^{\otimes k} + n^{-1/2} Z$ where every coordinate of $Z$ is drawn i.i.d. from $\mathcal{N}(0,1)$. We consider the negative log-likelihood:

$$L(\theta) = -\left\langle \theta^{\otimes k}, T \right\rangle.$$

The spherical gradient is given by:

$$b(\theta) = k P_\theta^\perp T[\theta^{\otimes k-1}].$$

### G.1  ODD $k$

**Lemma 27.** $\mathbb{E}_{z,Z} b(z) = c\theta^\star$ where $c = \Theta(d^{-\frac{k-1}{2}})$.

*Proof.* A direct calculation shows:

$$\mathbb{E}_{z,Z} b(z) = k\theta^\star \mathbb{E}_z \left[ (\theta^\star \cdot z)^{k-1} - (\theta^\star \cdot z)^{k+1} \right].$$

Note that $\theta^\star \cdot z$ is equal in distribution to $z_1$ so

$$c := k\mathbb{E}_z \left[ (\theta^\star \cdot z)^{k-1} - (\theta^\star \cdot z)^{k+1} \right]$$

is of order $\Theta(d^{-\frac{k-1}{2}})$. $\qquad\square$

Next, we will control concentrate the norm of the deviation from this population expectation.

**Lemma 28.** *With probability at least $1 - \delta$, we have the following:*

$$\|\mathbb{E}_z b(z) - \mathbb{E}_{z,Z} b(z)\| \lesssim \sqrt{\frac{d^{-(k-1)/2}(d + \log(1/\delta))}{n}}$$

*and in the $\theta^\star$ direction,*

$$|\theta^\star \cdot (\mathbb{E}_z b(z) - \mathbb{E}_{z,Z} b(z))| \lesssim \sqrt{\frac{d^{-(k-1)/2} \log(1/\delta)}{n}}$$

*Proof.* We first note that:

$$\mathbb{E}_z b(z) - \mathbb{E}_{z,Z} b(z) = k n^{-1/2} \mathbb{E}_z \left[ P_z^\perp Z[z^{\otimes k-1}] \right]$$

which can be seen to be a linear functional of the Gaussian tensor $Z$, as well as rotationally invariant by symmetry. Hence, we obtain that $\|\mathbb{E}_z b(z) - \mathbb{E}_{z,Z} b(z)\|^2 \overset{d}{=} \tau^2 \chi_d^2$, where $\tau^2 = \frac{1}{d} \mathbb{E}_Z \|\mathbb{E}_z b(z) - \mathbb{E}_{z,Z} b(z)\|^2$. This can be calculated as:

$$\begin{aligned}
\mathbb{E}_Z \|\mathbb{E}_z b(z) - \mathbb{E}_{z,Z} b(z)\|^2 &= \mathbb{E}_Z \left\| k n^{-1/2} \mathbb{E}_z \left[ P_z^\perp Z[z^{\otimes k-1}] \right] \right\|^2 \\
&\asymp n^{-1} \mathbb{E}_{z,z',Z} \left\langle P_z^\perp Z[z^{\otimes k-1}], P_{z'}^\perp Z[(z')^{\otimes k-1}] \right\rangle \\
&= n^{-1} \mathbb{E}_{z,z'} \left[ (z \cdot z')^{k-1} \left\langle P_z^\perp, P_{z'}^\perp \right\rangle \right] \\
&= n^{-1} \mathbb{E}_{z,z'} \left[ (z \cdot z')^{k-1} (d - 2 + (z \cdot z')^2) \right] \\
&\asymp n^{-1} d^{-(k-3)/2}
\end{aligned}$$

Therefore, $\tau^2 \asymp d^{-(k-1)/2}/n$. Finally, by $\chi_d^2$ concentration, we have that with probability at least $1 - \delta$, the magnitude of a $\chi_d^2$ random variable is bounded by $O(d + \sqrt{d \log(1/\delta)} + \log(1/\delta)) = O(d + \log(1/\delta))$ by the AM-GM inequality, and the result follows.

For the second equation, we note that $\theta^\star \cdot (\mathbb{E}_z b(z) - \mathbb{E}_{z,Z} b(z))$ is also a linear functional of the Gaussian tensor $Z$, so we simply have to consider its variance for concentration. Using the previous calculation, we obtain:

$$\mathrm{Var}_Z[\theta^\star \cdot (\mathbb{E}_z b(z) - \mathbb{E}_{z,Z} b(z))] = \mathbb{E}_Z \left| k n^{-1/2} \mathbb{E}_z \left[ P_z^\perp Z[z^{\otimes k-1}] \right] \cdot \theta^\star \right|^2 \asymp d^{-1} n^{-1} d^{-(k-3)/2}$$

where the $d^{-1}$ factor follows from the covariance being isotropic. This gives that with probability $1 - \delta$,

$$|\theta^\star \cdot (\mathbb{E}_z b(z) - \mathbb{E}_{z,Z} b(z))| \lesssim \sqrt{\frac{d^{-(k-1)/2} \log(1/\delta)}{n}}$$

$\square$

**Proposition 2.** *When $n \gtrsim d^{(k+1)/2}/\Delta^2$ for $\Delta \in (0, 1)$, it holds with probability $1 - e^{-d}$ that:*

$$\frac{\mathbb{E}_z b(z)}{\|\mathbb{E}_z b(z)\|} \cdot \theta^\star \geq 1 - \Delta$$

*Moreover, when $n \gtrsim d^{k/2}$, it holds with probability $1 - e^{-d^c}$ for $c < 1/2$ that:*

$$\frac{\mathbb{E}_z b(z)}{\|\mathbb{E}_z b(z)\|} \cdot \theta^\star \gtrsim d^{-1/4}$$

*Proof.* When $n \gtrsim d^{(k+1)/2}/\Delta^2$, we have with probability $1 - e^{-d}$ that:

$$\begin{aligned}
\frac{\mathbb{E}_z b(z) \cdot \theta^\star}{\|\mathbb{E}_z b(z)\|} &\geq \frac{\mathbb{E}_{Z,z} b(z) \cdot \theta^\star - |(\mathbb{E}_z b(z) - \mathbb{E}_{Z,z} b(z)) \cdot \theta^\star|}{\|\mathbb{E}_{z,Z} b(z)\| + \|\mathbb{E}_z b(z) - \mathbb{E}_{z,Z} b(z)\|} \\
&\geq \frac{d^{-(k-1)/2}(1 - \Delta/2)}{d^{-(k-1)/2}(1 + \Delta/2)} \\
&\geq 1 - \Delta
\end{aligned}$$

as desired. When $n \gtrsim d^{k/2}$, we have with probability $1 - e^{-d^c}$ that:

$$
\begin{aligned}
\frac{\mathbb{E}_z b(z) \cdot \theta^\star}{\|\mathbb{E}_z b(z)\|} &\geq \frac{\mathbb{E}_{Z,z} b(z) \cdot \theta^\star - |(\mathbb{E}_z b(z) - \mathbb{E}_{Z,z} b(z)) \cdot \theta^\star|}{\|E_{z,Z} b(z)\| + \|\mathbb{E}_z b(z) - \mathbb{E}_{z,Z} b(z)\|} \\
&\gtrsim \frac{d^{-(k-1)/2}}{d^{-(k-1)/2}(1 + d^{1/4})} \\
&\gtrsim d^{-1/4}
\end{aligned}
$$

where in the second inequality we use the fact that $|(\mathbb{E}_z b(z) - \mathbb{E}_{Z,z} b(z)) \cdot \theta^\star| \ll \mathbb{E}_{Z,z} b(z) \cdot \theta^\star$ due to $c < 1/2$. $\qquad\square$

## G.2 EVEN $k$

In this section, our goal is to concentrate the self-adjoint random matrix $G := \mathbb{E}_z[zb(z)^\top + b(z)z^\top]$.

**Lemma 29.** $\mathbb{E}_{z,Z}[G] = \Theta(d^{-k/2})\theta^\star\theta^{\star\top} - \Theta(d^{-(k+2)/2})P_{\theta^\star}^\perp$

*Proof.* We have that:

$$
\mathbb{E}_{z,Z}[zb(z)^\top] = k\mathbb{E}_z[z\theta^{\star\top}(\theta^\star \cdot z)^{k-1} - (\theta^\star \cdot z)^k zz^\top]
$$

By symmetry, it holds that:

$$
\mathbb{E}_z[z\theta^{\star\top}(\theta^\star \cdot z)^{k-1} - (\theta^\star \cdot z)^k zz^\top] = \Theta(d^{-k/2})\theta^\star\theta^{\star\top} - \Theta(d^{-(k+2)/2})P_{\theta^\star}^\perp
$$

Similar calculations hold for $\mathbb{E}_{z,Z}[b(z)z^\top]$ as it is just the transpose, and the result follows. $\qquad\square$

**Lemma 30.** *With probability at least $1 - \delta$, it holds that:*

$$
\|G - \mathbb{E}_Z G\|_2 \lesssim \sqrt{\frac{d^{-(k+2)/2}(d + \log(1/\delta))}{n}}
$$

*Proof.* Note that $G - \mathbb{E}_Z G$ is self-adjoint. Therefore, it holds that:

$$
\begin{aligned}
\|G - \mathbb{E}_Z G\|_2 &\leq 2 \sup_{v \in \mathcal{N}_{1/4}} |v^\top(G - \mathbb{E}_Z G)v| \\
&\leq 2 \sup_{v \in \mathcal{N}_{1/4}} |v^\top(\mathbb{E}_z[zb(z)^\top] - \mathbb{E}_{z,Z}[zb(z)^\top])v| + 2 \sup_{v \in \mathcal{N}_{1/4}} |v^\top(\mathbb{E}_z[b(z)z^\top] - \mathbb{E}_{z,Z}[b(z)z^\top])v|
\end{aligned}
$$

where $\mathcal{N}_{1/4}$ denotes a $1/4$-net of $S^{d-1}$. It now suffices to bound each of these two terms individually.

Let us start with the first term. Consider for a fixed $v \in S^{d-1}$, the quantity

$$
v^\top[\mathbb{E}_z[zb(z)^\top] - \mathbb{E}_{z,Z}[zb(z)^\top]]v = kn^{-1/2} \cdot v^\top \mathbb{E}_z[z(P_z^\perp Z[z^{\otimes k-1}])^\top]v
$$

Since this quantity is a linear functional of a Gaussian tensor $Z$, it suffices to analyze just the variance to obtain a concentration.

$$
\begin{aligned}
&\mathrm{Var}_Z\big[v^\top[\mathbb{E}_z[zb(z)^\top] - \mathbb{E}_{z,Z}[zb(z)^\top]]v\big] \\
&\lesssim n^{-1}\mathbb{E}_Z\big[\mathbb{E}_z[v^\top[z(P_z^\perp Z[z^{\otimes k-1}])^\top]v]^2\big] \\
&= n^{-1}\mathbb{E}_Z\big[\mathbb{E}_{z,z'}[(v \cdot z)(v \cdot z')[(P_z^\perp Z[z^{\otimes k-1}])^\top v][(P_{z'}^\perp Z[z'^{\otimes k-1}])^\top v]]\big] \\
&= n^{-1}\mathbb{E}_{z,z'}\big[(v \cdot z)(v \cdot z')\mathbb{E}_Z[(P_z^\perp Z[z^{\otimes k-1}])^\top v(P_{z'}^\perp Z[z'^{\otimes k-1}])^\top v]]\big] \\
&= n^{-1}\mathbb{E}_{z,z'}\big[(v \cdot z)(v \cdot z') \cdot v^\top P_z^\perp \mathbb{E}_Z[(Z[z^{\otimes k-1}])(Z[z'^{\otimes k-1}])^\top]P_{z'}^\perp v\big] \\
&= n^{-1}\mathbb{E}_{z,z'}\big[(v \cdot z)(v \cdot z') \cdot v^\top P_z^\perp (z \cdot z')^{k-1}I_d P_{z'}^\perp v\big] \\
&= n^{-1}\mathbb{E}_{z,z'}\big[(v \cdot z)(v \cdot z')(z \cdot z')^{k-1}\big] \\
&\quad - n^{-1}\mathbb{E}_{z,z'}\big[(v \cdot z)^3(v \cdot z')(z \cdot z')^{k-1}]\big] - n^{-1}\mathbb{E}_{z,z'}\big[(v \cdot z)(v \cdot z')^3(z \cdot z')^{k-1}\big] \\
&\quad + n^{-1}\mathbb{E}_{z,z'}\big[(v \cdot z)^2(v \cdot z')^2(z \cdot z')^k\big]
\end{aligned}
$$

In the last expression, the first term is the main term, and the latter three are due to the at least one of the projection terms. For the main term, we can bound it by $\Theta(d^{-(k+2)/2}/n)$. For the latter three, we have that they are $O(d^{-(k+4)/2}/n)$. Hence, the entire variance expression is $\Theta(d^{-(k+2)/2}/n)$.

Therefore, we have that for an arbitrary $v \in S^{d-1}$, it holds with probabillity at least $1 - \delta/9^d$:

$$\left| v^\top \left[ \mathbb{E}_z[zb(z)^\top] - \mathbb{E}_{z,Z}[zb(z)^\top] \right] v \right| \lesssim \sqrt{\frac{d^{-(k+2)/2} \log(9^d/\delta)}{n}} \asymp \sqrt{\frac{d^{-(k+2)/2}(d + \log(1/\delta))}{n}}$$

We now consider a $1/4$-net $\mathcal{N}_{1/4}$ over $S^{d-1}$, which has size at most $9^d$. Union bounding over $v \in \mathcal{N}_{1/4}$, we have that with probability at least $1 - \delta$ that:

$$\sup_{v \in \mathcal{N}_{1/4}} \left| v^\top \left[ \mathbb{E}_z[zb(z)^\top] - \mathbb{E}_{z,Z}[zb(z)^\top] \right] v \right| \lesssim \sqrt{\frac{d^{-(k+2)/2}(d + \log(1/\delta))}{n}}$$

By a similar argument, we obtain that:

$$\sup_{v \in \mathcal{N}_{1/4}} \left| v^\top \left[ \mathbb{E}_z[b(z)z^\top] - \mathbb{E}_{z,Z}[b(z)z^\top] \right] v \right| \lesssim \sqrt{\frac{d^{-(k+2)/2}(d + \log(1/\delta))}{n}}$$

Adding these yields the desired result. $\qquad\square$

**Proposition 3.** *When $n \gtrsim d^{k/2}/\Delta^2$ for $\Delta \in (0,1)$, it holds with probability at least $1 - e^{-d}$ that the top eigenvector $v$ of $\mathbb{E}_z[zb(z)^\top + b(z)z^\top]$ satisfies $(v \cdot \theta^\star)^2 \geq 1 - \Delta$.*

*Proof.* First, note that the eigengap of the expectation over $Z$ is $\Theta(d^{-k/2})$. From the previous lemma, we know that with probability $1 - e^{-d}$, $\|G - \mathbb{E}_Z G\|_2 \leq d^{-k/2}\Delta/2$ for our choice of $n \gtrsim d^{k/2}/\Delta^2$ (with an appropriately chosen constant). Hence, the eigengap of $G$ is bounded below by $d^{-k/2}(1 - \Delta/2)$. From the Davis-Kahan theorem, we have that

$$\sqrt{1 - (v \cdot \theta^\star)^2} = \sin \angle(v, \theta^\star) \leq \frac{d^{-k/2}\Delta/2}{d^{-k/2}(1 - \Delta/2)} \leq \Delta$$

Rearranging yields the corollary.

$\qquad\square$

### G.3 LIPSCHITZNESS OF $b$

It remains to show that $b$ is bounded and Lipschitz, which is formalized through the next two lemmas.

**Lemma 31.** *With probability $1 - e^{-d}$, it holds that:*

$$\sup_\theta \|b(\theta)\| \lesssim 1 + \sqrt{d/n}$$

*Proof.* Observe that with probability at least $1 - e^{-cd}$,

$$\sup_\theta \|b(\theta)\| \lesssim 1 + n^{-1/2} \sup_\theta Z[\theta^{\otimes k-1}] \leq 1 + n^{-1/2}\|Z\|_{op} \lesssim 1 + \sqrt{d/n}$$

where the operator norm bound on $Z$ follows from a standard covering argument. $\qquad\square$

**Lemma 32.** *In the same setting as Lemma 31,*

$$\|b(\theta) - b(\theta')\| \lesssim (1 + \sqrt{d/n})\|\theta - \theta'\|$$

*Proof.*

$$\begin{aligned}
\|b(\theta) - b(\theta')\| &\leq k\left\| P_\theta^\perp T[\theta^{\otimes k-1}] - P_{\theta'}^\perp T[(\theta')^{\otimes k-1}] \right\| \\
&\leq k\left\| (P_\theta^\perp - P_{\theta'}^\perp)T[\theta^{\otimes k-1}] + P_{\theta'}^\perp(T[\theta^{\otimes k-1} - (\theta')^{\otimes k-1}]) \right\| \\
&\lesssim (1 + \sqrt{d/n})\|\theta - \theta'\|
\end{aligned}$$

where the inequality for the second term follows from the fact that if $\theta' = \theta + E$:

$$\left\| T[(\theta + E)^{\otimes k-1} - \theta^{\otimes k-1}] \right\| = \sum_{j=1}^{k-1} \binom{k-1}{j} T[E^{\otimes j} \otimes \theta^{\otimes k-1-j}] \leq \|T\|_{op} \sum_{j=1}^{k-1} \|E\|^j \lesssim \|T\|_{op} \|E\|.$$

$\square$

## H  SINGLE INDEX MODELS

Recall that by assumption, our activation satisfies $\sup_z \sigma^{(k)}(z) = O(1)$ for $k = 0, 1, 2$. Define $b_i(\theta)$ to be the negative spherical gradient on the $i$th datapoint:

$$b_i(\theta) := y_i P_\theta^\perp x_i \sigma'(\theta \cdot x_i).$$

We will use $\mathbb{E}_i$ to denote the expectation with respect to the data. We will also let $z \sim \mathrm{Unif}(S^{d-1})$.

### H.1  ODD $k^\star$

**Lemma 33.** $\mathbb{E}_{i,z} b_i(z) = c\theta^\star$ where $c = \Theta(d^{-\frac{k^\star-1}{2}})$.

*Proof.* First note that by Hermite expanding $y$ and $\sigma$ we have that:

$$\mathbb{E}_i y_i \sigma(z \cdot x_i) = \mathbb{E}_i[\sigma(\theta^\star \cdot x)\sigma(z \cdot x)] = \sum_{k \geq k^\star} c_k^2 (z \cdot \theta^\star)^k.$$

Taking a spherical gradient with respect to $\theta$ gives:

$$\mathbb{E}_i b_i(z) = \sum_{k \geq k^\star} k c_k^2 (P_z^\perp \theta^\star)(z \cdot \theta^\star)^{k-1}.$$

We can now average over the sphere. First by (Damian et al., 2023, Lemma 26),

$$\mathbb{E}_z \sum_{k \geq k^\star} k c_k^2 (z \cdot \theta^\star)^{k-1} \lesssim d^{-\frac{k^\star-1}{2}}.$$

In addition by isolating the $k = k^\star$ term, it is at least order $d^{-\frac{k^\star-1}{2}}$. Next we handle the projection term:

$$\mathbb{E}_z \sum_{k \geq k^\star} k c_k^2 z (z \cdot \theta^\star)^k = \theta^\star \mathbb{E}_z \sum_{k \geq k^\star} k c_k^2 (z \cdot \theta^\star)^{k+1}$$

and this is upper bounded by $\Theta\left(d^{-\frac{k^\star+1}{2}}\right)$ which completes the proof. $\square$

**Lemma 34.** *With probability $1 - \delta$, we have that:*

$$\|\mathbb{E}_z b(z) - \mathbb{E}_{i,z} b(z)\| \lesssim \sqrt{\frac{d^{-(k^\star-1)/2}(d + \log \frac{1}{\delta})}{n}} + \frac{\sqrt{d + \log n} \log \frac{1}{\delta}}{n} \tag{3}$$

*and in the $\theta^\star$ direction,*

$$|\theta^\star \cdot (\mathbb{E}_z b(z) - \mathbb{E}_{i,z} b(z))| \lesssim \sqrt{\frac{d^{-(k^\star-1)/2} \log \frac{1}{\delta}}{n}} + \frac{\sqrt{\log n} \log \frac{1}{\delta}}{n}$$

*Proof.* We can decompose:

$$\mathbb{E}_z b_i(z) = y_i x_i \mathbb{E}_z \sigma'(z \cdot x_i) - y_i \mathbb{E}_z[z(z \cdot x_i)\sigma'(z \cdot x_i)].$$

We first concentrate in the direction of $\theta^\star$. We will analyze the main term and the projection term separately. For the main term, we have that:

$$
\begin{aligned}
\mathrm{Var}_x\left[\frac{1}{n}\sum_i y_i(x_i\cdot\theta^\star)\mathbb{E}_z\sigma'(z\cdot x_i)\right] &= \frac{1}{n}\mathrm{Var}_x[y(x\cdot\theta^\star)\mathbb{E}_z\sigma'(z\cdot x)] \\
&\leq \frac{1}{n}\mathbb{E}_x\left[y^2(x\cdot\theta^\star)^2\mathbb{E}_{z,z'}[\sigma'(z\cdot x)\sigma'(z'\cdot x)]\right] \\
&\lesssim \frac{1}{n}\mathbb{E}_x\left[(x\cdot\theta^\star)^2\mathbb{E}_{z,z'}[\sigma'(z\cdot x)\sigma'(z'\cdot x)]\right] \\
&= \frac{1}{n}\mathbb{E}_x\left[\left(\frac{x}{\|x\|}\cdot\theta^\star\right)^2\right]\cdot\mathbb{E}_x\left[\|x\|^2\mathbb{E}_{z,z'}[\sigma'(z\cdot x)\sigma'(z'\cdot x)]\right] \\
&\lesssim \frac{1}{n}\cdot\frac{1}{d}\cdot d^{-(k^\star-3)/2} \\
&\lesssim \frac{d^{-(k^\star-1)/2}}{n}
\end{aligned}
$$

where in the third to last line we used the polar decomposition of $x$, and in the second to last line we used the fact that $\mathbb{E}_x\left[\left(\frac{x}{\|x\|}\cdot\theta^\star\right)^2\right]=\Theta(1/d)$ and:

$$
\begin{aligned}
&\mathbb{E}_x\left[\|x\|^2\mathbb{E}_{z,z'}[\sigma'(z\cdot x)\sigma'(z'\cdot x)]\right] \\
&= \mathbb{E}_x\left[\mathbf{1}_{\|x\|^2\leq Cd}\|x\|^2\mathbb{E}_{z,z'}[\sigma'(z\cdot x)\sigma'(z'\cdot x)]\right] + \mathbb{E}_x\left[\mathbf{1}_{\|x\|^2\geq Cd}\|x\|^2\mathbb{E}_{z,z'}[\sigma'(z\cdot x)\sigma'(z'\cdot x)]\right] \\
&\lesssim d\mathbb{E}_x[\mathbb{E}_{z,z'}[\sigma'(z\cdot x)\sigma'(z'\cdot x)]] + \sqrt{\mathbb{P}[\|x\|^2\geq Cd]\mathbb{E}_x\left[(\|x\|^2\mathbb{E}_{z,z'}[\sigma'(z\cdot x)\sigma'(z'\cdot x)])^2\right]} \\
&\lesssim d^{-(k^\star-3)/2}
\end{aligned}
$$

which follows from $\sigma'$ having information exponent $k^\star-1$, and $\mathbb{P}[\|x\|^2\geq Cd]$ being exponentially small for $C>1$ by standard $\chi^2$ concentration.

We also note that $y_i(x_i\cdot\theta^\star)\mathbb{E}_z\sigma'(z\cdot x_i)$ is $O(1)$-subgaussian. Therefore, by Lemma 25, it holds with probability $1-\delta$ that:

$$
\left|\frac{1}{n}\sum_i y_i(x_i\cdot\theta^\star)\mathbb{E}_z\sigma'(z\cdot x_i) - \mathbb{E}_x\left[\frac{1}{n}\sum_i y_i(x_i\cdot\theta^\star)\mathbb{E}_z\sigma'(z\cdot x_i)\right]\right| \lesssim \sqrt{\frac{d^{-(k^\star-1)/2}\log(1/\delta)}{n}} + \frac{\sqrt{\log n}\log(1/\delta)}{n}
$$

For the projection term in the direction of $\theta^\star$, we have that:

$$
\begin{aligned}
\mathrm{Var}_x\left[\frac{1}{n}\sum_i y_i\mathbb{E}_z[(z\cdot\theta^\star)(z\cdot x_i)\sigma'(z\cdot x_i)]\right] &= \frac{1}{n}\mathrm{Var}_x[y\mathbb{E}_z[(z\cdot\theta^\star)(z\cdot x)\sigma'(z\cdot x)]] \\
&= \frac{1}{n}\mathrm{Var}_x\left[y\frac{x\cdot\theta^\star}{\|x\|^2}\mathbb{E}_z\left[(z\cdot x)^2\sigma'(z\cdot x)\right]\right] \\
&\leq \frac{1}{n}\mathbb{E}_x\left[y^2\frac{(x\cdot\theta^\star)^2}{\|x\|^4}\mathbb{E}_z\left[(z\cdot x)^2\sigma'(z\cdot x)\right]^2\right] \\
&\lesssim \frac{1}{n}\mathbb{E}_x\left[\left(\frac{x}{\|x\|}\cdot\theta^\star\right)^2\right]\mathbb{E}_x\left[\frac{\mathbb{E}_z\left[(z\cdot x)^2\sigma'(z\cdot x)\right]^2}{\|x\|^2}\right] \\
&\lesssim \frac{1}{n}\cdot\frac{1}{d}\cdot\frac{d^{-(k^\star-3)/2}}{d} \\
&= \frac{d^{-(k^\star+1)/2}}{n}
\end{aligned}
$$

where the second to last line follows from:

$$\mathbb{E}_x\left[\frac{\mathbb{E}_z\left[(z\cdot x)^2\sigma'(z\cdot x)\right]^2}{\|x\|^2}\right] = \mathbb{E}_x\left[\mathbf{1}_{\|x\|^2\geq Cd}\frac{\mathbb{E}_z\left[(z\cdot x)^2\sigma'(z\cdot x)\right]^2}{\|x\|^2}\right] + \mathbb{E}_x\left[\mathbf{1}_{\|x\|^2\leq Cd}\frac{\mathbb{E}_z\left[(z\cdot x)^2\sigma'(z\cdot x)\right]^2}{\|x\|^2}\right]$$

$$\leq \frac{1}{d}\mathbb{E}_x\left[\mathbb{E}_z[(z\cdot x)^2\sigma'(z\cdot x)]^2\right] + \sqrt{\mathbb{P}[\mathbf{1}_{\|x\|^2\leq Cd}]\cdot\mathbb{E}_x\left[\left(\frac{\mathbb{E}_z[(z\cdot x)^2\sigma'(z\cdot x)]^2}{\|x\|^2}\right)^2\right]}$$

$$\lesssim \frac{1}{d}\cdot d^{-(k^\star-3)/2}$$

since $t^2\sigma'(t)$ has information exponent $k^\star - 3$ by Lemma 17 and $\mathbb{P}[\mathbf{1}_{\|x\|^2\leq Cd}]$ is exponentially small for $C < 1$. Moreover, we can see that this is $O(1/d)$-subgaussian:

$$|y\mathbb{E}_z[(z\cdot\theta^\star)(z\cdot x)\sigma'(z\cdot x)]| = \left|y\frac{x\cdot\theta^\star}{\|x\|^2}\mathbb{E}_z\left[(z\cdot x)^2\sigma'(z\cdot x)\right]\right|$$

$$\lesssim \left|(x\cdot\theta^\star)\mathbb{E}_z\left[\left(z\cdot\frac{x}{\|x\|}\right)^2\sigma'(z\cdot x)\right]\right|$$

$$\leq \left|(x\cdot\theta^\star)\sqrt{\mathbb{E}_z\left[\left(z\cdot\frac{x}{\|x\|}\right)^4\right]\mathbb{E}_z[\sigma'(z\cdot x)^2]}\right|$$

$$\lesssim \left|(x\cdot\theta^\star)\cdot\frac{1}{d}\right|$$

Therefore by Lemma 25, with probability $1-\delta$ it holds that:

$$\left|\frac{1}{n}\sum_i y_i\mathbb{E}_z[(z\cdot\theta^\star)(z\cdot x_i)\sigma'(z\cdot x_i)] - \mathbb{E}_x[y\mathbb{E}_z[(z\cdot\theta^\star)(z\cdot x)\sigma'(z\cdot x)]]\right| \lesssim \sqrt{\frac{d^{-(k^\star+1)/2}\log(1/\delta)}{n}} + \frac{\sqrt{\log n}\log(1/\delta)}{dn}$$

Altogether combining the main and projection term in the $\theta^\star$ direction, we have that with probability $1-\delta$ that:

$$|\theta^\star\cdot(\mathbb{E}_z b(z) - \mathbb{E}_{i,z}b(z))| \lesssim \sqrt{\frac{d^{-(k^\star-1)/2}\log(1/\delta)}{n}} + \frac{\sqrt{\log n}\log(1/\delta)}{n}$$

We now concentrate the entire norm of $\mathbb{E}_z b(z) - \mathbb{E}_{i,z}b(z)$, and once again we will handle the main and projection term separately. By the same variance and subgaussian calculations as before, we can apply Lemma 26 to obtain that with probability $1-\delta$,

$$\|\mathbb{E}_z b(z) - \mathbb{E}_{i,z}b(z)\| \lesssim \sqrt{\frac{d^{-(k^\star-1)/2}(d+\log\frac{1}{\delta})}{n}} + \frac{\sqrt{d+\log n}\log\frac{1}{\delta}}{n}$$

The desired result follows. $\qquad\square$

**Proposition 4.** *When $n\gtrsim d^{(k^\star+1)/2}/\Delta^2$ for $\Delta\in(0,1)$, it holds with probability $1-e^{-d^c}$ that:*

$$\frac{\mathbb{E}_z b(z)}{\|\mathbb{E}_z b(z)\|}\cdot\theta^\star \geq 1-\Delta$$

*Moreover, when $n\gtrsim d^{k^\star/2}$, it holds with probability $1-e^{-d^c}$ that:*

$$\frac{\mathbb{E}_z b(z)}{\|\mathbb{E}_z b(z)\|}\cdot\theta^\star \gtrsim d^{-1/4}$$

*Proof.* When $n\gtrsim d^{(k^\star+1)/2}/\Delta^2$, we have with probability $1-e^{-d^c}$ that:

$$\frac{\mathbb{E}_z b(z)\cdot\theta^\star}{\|\mathbb{E}_z b(z)\|} \geq \frac{\mathbb{E}_{i,z}b(z)\cdot\theta^\star - |(\mathbb{E}_z b(z) - \mathbb{E}_{i,z}b(z))\cdot\theta^\star|}{\|\mathbb{E}_{i,z}b(z)\| + \|\mathbb{E}_z b(z) - \mathbb{E}_{i,z}b(z)\|}$$

$$\geq \frac{d^{-(k^\star-1)/2}(1-\Delta/2)}{d^{-(k^\star-1)/2}(1+\Delta/2)}$$

$$\geq 1-\Delta$$

as desired. When $n \gtrsim d^{k^\star/2}$, we have with probability $1 - e^{-d^c}$ that:

$$
\frac{\mathbb{E}_z b(z) \cdot \theta^\star}{\|\mathbb{E}_z b(z)\|} \geq \frac{\mathbb{E}_{i,z} b(z) \cdot \theta^\star - |(\mathbb{E}_z b(z) - \mathbb{E}_{i,z} b(z)) \cdot \theta^\star|}{\|E_{i,z} b(z)\| + \|\mathbb{E}_z b(z) - \mathbb{E}_{i,z} b(z)\|}
$$
$$
\gtrsim \frac{d^{-(k^\star - 1)/2}}{d^{-(k^\star - 1)/2}(1 + d^{1/4})}
$$
$$
\gtrsim d^{-1/4}
$$

where in the second inequality we use the fact that the first term of the numerator dominates for $c < 1/4$, and the second term in the denominator is of order $d^{-(k^\star - 1)/2} \cdot d^{1/4}$ for this same choice of $c$. $\qquad\square$

## H.2 Even $k^\star$

**Lemma 35.** $\mathbb{E}_{i,z}[zb(z)^\top] = c\theta^\star \theta^{\star\top} - gP_{\theta^\star}^\perp$ where $c = \Theta(d^{-k^\star/2})$ and $g = O(d^{-(k^\star+2)/2})$.

*Proof.* We will fix $z$ first and then take average over the sphere of $z$. First,

$$
\mathbb{E}_i[zx_i^\top \sigma(\theta^\star \cdot x_i)\sigma'(z \cdot x_i)P_z^\perp] = z\mathbb{E}_i[x_i^\top \sigma(\theta^\star \cdot x_i)\sigma'(z \cdot x_i)] - z\mathbb{E}_i[x_i^\top \sigma(\theta^\star \cdot x_i)\sigma'(z \cdot x_i)]zz^\top
$$

Let $c_i$ be the Hermite coefficients for $\sigma$. For the first term, we have by Stein's lemma that:

$$
z\mathbb{E}_i[x_i^\top \sigma(\theta^\star \cdot x_i)\sigma'(z \cdot x_i)] = z\mathbb{E}_i[\sigma'(\theta^\star \cdot x_i)\sigma'(z \cdot x)]\theta^{\star\top} + \mathbb{E}_i[\sigma(\theta^\star \cdot x_i)\sigma''(z \cdot x_i)]zz^\top
$$
$$
= z\sum_{k \geq k^\star - 1} c_k^2 (\theta^\star \cdot z)^k \theta^{\star\top} + \sum_{k \geq k^\star}(k+2)(k+1)c_k c_{k+2}(\theta^\star \cdot z)^k zz^\top
$$

We now proceed to handle the projection term:

$$
z\mathbb{E}_i[x_i^\top \sigma(\theta^\star \cdot x_i)\sigma'(z \cdot x_i)]zz^\top = z\sum_{k \geq k^\star - 1} c_k^2 (\theta^\star \cdot z)^k \theta^{\star\top} zz^\top + \sum_{k \geq k^\star}(k+2)(k+1)c_k c_{k+2}(\theta^\star \cdot z)^k zz^\top zz^\top
$$
$$
= z\sum_{k \geq k^\star - 1} c_k^2 (\theta^\star \cdot z)^{k+1} z^\top + \sum_{k \geq k^\star}(k+2)(k+1)c_k c_{k+2}(\theta^\star \cdot z)^k zz^\top
$$

Therefore, after combining and before taking expectation over $z$, our expression is:

$$
z\sum_{k \geq k^\star - 1} c_k^2 (\theta^\star \cdot z)^k \theta^{\star\top} - z\sum_{k \geq k^\star - 1} c_k^2 (\theta^\star \cdot z)^{k+1} z^\top
$$

We now take expectation of $z$ over the sphere. For the first term, we have that

$$
\mathbb{E}_z\left[z\sum_{k \geq k^\star - 1} c_k^2 (\theta^\star \cdot z)^k\right]\theta^\star = \sum_{j \geq 0} \Theta(d^{-(k^\star+2j)/2})\theta^\star \theta^{\star\top} = \Theta(d^{-k^\star/2})\theta^\star \theta^{\star\top}
$$

For the second term, we have that

$$
\mathbb{E}_z\left[\sum_{k \geq k^\star - 1} c_k^2 (\theta^\star \cdot z)^{k+1} zz^\top\right] = \Theta(d^{-(k^\star+2)/2})\theta^\star \theta^{\star\top} + \Theta(d^{-(k^\star+2)/2})P_{\theta^\star}^\perp
$$

where the two $\Theta$ hide different absolute constants. Nonetheless, the main part of our desired expression is $\Theta(d^{-k^\star/2})\theta^\star \theta^{\star\top}$, and this gives the desired result. $\qquad\square$

**Corollary 4.** $\mathbb{E}_{i,z}[zb(z)^\top + b(z)z^\top] = c\theta^\star \theta^{\star\top} + gP_{\theta^\star}^\perp$ where $c = \Theta(d^{-k^\star/2})$ and $g = O(d^{-(k^\star+2)/2})$.

*Proof.* This follows directly from the fact that we are just adding the transpose. $\qquad\square$

In the rest of the section, our goal is to concentrate the self-adjoint random matrix $G := \mathbb{E}_z[zb(z)^\top + b(z)z^\top]$; however, for the sake of exposition we will simply consider $\mathbb{E}_z[zb(z)^\top]$, and we will note, when necessary, the properties that transition to the case of $G$.

**Lemma 36.** *With probability at least $1 - \delta$, it holds that:*

$$\left\| \frac{1}{n} \sum_{i=1}^{n} \mathbb{E}_z[zy_i\sigma'(z \cdot x_i)x_i^\top P_z^\perp] - \mathbb{E}_{i,z}[zy_i\sigma'(z \cdot x_i)x_i^\top P_z^\perp] \right\|_2$$

$$\lesssim \sqrt{\frac{d^{-(k^\star+2)/2}(d + \log(1/\delta))}{n}} + \frac{d + \log(1/\delta)}{dn}$$

*Proof.* We will now concentrate $\frac{1}{n}\sum_i \mathbb{E}_z[zb(z)^\top] = \frac{1}{n}\sum_{i=1}^n \mathbb{E}_z[zy_i\sigma'(z \cdot x_i)x_i^\top P_z^\perp] - \mathbb{E}_{i,z}[zb(z)^\top]$ in operator norm. By the epsilon-net bound on the operator norm, it suffices to consider for an arbitrary $v \in S^{d-1}$ the quantity $v^\top \left[\frac{1}{n}\sum_{i=1}^n \mathbb{E}_z[zy_i\sigma'(z \cdot x_i)x_i^\top P_z^\perp] - \mathbb{E}_{i,z}[zb(z)^\top]\right]v$. First, note that the projection term gives the following decomposition:

$$\mathbb{E}_z[zy_i\sigma'(z \cdot x_i)x_i^\top P_z^\perp] = y_i\mathbb{E}_z[z\sigma'(z \cdot x_i)x_i^\top] - y_i\mathbb{E}_z[z\sigma'(z \cdot x_i)(z \cdot x_i)z^\top]$$

We will handle the two terms separately. For the first term, the variance is bounded above by:

$$\begin{aligned}
\frac{1}{n}\mathbb{E}_i\left[\mathbb{E}_z[(v \cdot z)y\sigma'(z \cdot x)(x \cdot v)]^2\right] &= \frac{1}{n}\mathbb{E}_i\left[y^2(x \cdot v)^2\mathbb{E}_z[(v \cdot z)\sigma'(z \cdot x)]^2\right] \\
&\lesssim \frac{1}{n}\mathbb{E}_i\left[(x \cdot v)^2\mathbb{E}_z[(v \cdot z)\sigma'(z \cdot x)]^2\right] \\
&= \frac{1}{n}\mathbb{E}_i\left[\frac{(x \cdot v)^4}{\|x\|^4}\mathbb{E}_z[(x \cdot z)\sigma'(z \cdot x)]^2\right] \\
&= \frac{1}{n}\mathbb{E}_i\left[\frac{(x \cdot v)^4}{\|x\|^4}\right] \cdot \mathbb{E}_i\left[\mathbb{E}_z[(x \cdot z)\sigma'(x \cdot z)]^2\right] \\
&\lesssim \frac{1}{n} \cdot \frac{1}{d^2} \cdot d^{-(k^\star-2)/2} \\
&= \frac{d^{-(k^\star+2)/2}}{n}
\end{aligned}$$

where in the third line we use the fact that by symmetry:

$$\mathbb{E}_z[(v \cdot z)\sigma'(z \cdot x)] = \frac{v \cdot x}{\|x\|^2}\mathbb{E}_z[(z \cdot x)\sigma'(z \cdot x)]$$

In the fourth line we use the independence between the direction of $x$ and the norm of $x$, and in the second to last line the fact that the information exponent of $t \cdot \sigma'(t)$ is $k^\star - 2$.

In addition, we will also show the term itself is $O(1/d)$-exponential, from which we will combine with the variance bound via Bernstein. Rewriting the term, we have:

$$y_i(x_i \cdot v)\mathbb{E}_z[(v \cdot z)\sigma'(z \cdot x_i)] = y_i(x_i \cdot v)\frac{v \cdot x_i}{\|x_i\|^2}\mathbb{E}_z[(z \cdot x_i)\sigma'(z \cdot x_i)]$$

Since $|y_i| = O(1)$ and $x_i \cdot v$ is $O(1)$ subgaussian, it suffices to show that $\frac{v \cdot x_i}{\|x_i\|^2}\mathbb{E}_z[(z \cdot x_i)\sigma'(z \cdot x_i)]$ is $O(1/d)$ subgaussian. This follows from:

$$\begin{aligned}
\left|\frac{v \cdot x_i}{\|x_i\|^2}\mathbb{E}_z[(z \cdot x_i)\sigma'(z \cdot x_i)]\right| &= \left|\frac{v \cdot x_i}{\|x_i\|^2}\mathbb{E}_z[(z \cdot x_i)(\sigma'(0) + [\sigma'(z \cdot x_i) - \sigma'(0)])]\right| \\
&= \left|\frac{v \cdot x_i}{\|x_i\|^2}\mathbb{E}_z[(z \cdot x_i)(\sigma'(z \cdot x_i) - \sigma'(0))]\right| \\
&\lesssim \frac{|v \cdot x_i|}{\|x_i\|^2}\mathbb{E}_z[(z \cdot x_i)^2] \\
&= \frac{|v \cdot x_i|}{d}
\end{aligned}$$

Since this is upper bounded by $O(1/d)$ times a half-Gaussian, we have that this is $O(1/d)$ subgaussian. Therefore, the entire term is $O(1/d)$ subexponential. In particular, for a fixed $v \in S^{d-1}$, Bernstein's inequality gives that with probability at least $1 - \delta/9^d$:

$$\left\| v^\top \left[\frac{1}{n}\sum_{i=1}^n \mathbb{E}_z[zy_i\sigma'(z \cdot x_i)x_i^\top] - \mathbb{E}_{i,z}[zy_i\sigma'(z \cdot x_i)x_i^\top]\right]v \right\| \lesssim \sqrt{\frac{d^{-(k^\star+2)/2}\log(9^d/\delta)}{n}} + \frac{\log(9^d/\delta)}{dn}$$

We now handle the projection term. Here, the variance is bounded above by:

$$\frac{1}{n}\mathbb{E}_i[\mathbb{E}_z[(v\cdot z)^2 y\sigma'(z\cdot x)(z\cdot x)]^2] = \frac{1}{n}\mathbb{E}_i[y^2\mathbb{E}_z[(v\cdot z)^2\sigma'(z\cdot x)(z\cdot x)]^2]$$

$$\lesssim \frac{1}{n}\mathbb{E}_i[\mathbb{E}_z[(v\cdot z)^2\sigma'(z\cdot x)(z\cdot x)]^2]$$

$$= \frac{1}{n}\mathbb{E}_i[\mathbb{E}_{z,z'}[(v\cdot z)^2(v\cdot z')^2\sigma'(z\cdot x)(z\cdot x)\sigma'(z'\cdot x)(z'\cdot x)]]$$

$$= \frac{1}{n}\mathbb{E}_{z,z'}[(v\cdot z)^2(v\cdot z')^2\mathbb{E}_i[\sigma'(z\cdot x)(z\cdot x)\sigma'(z'\cdot x)(z'\cdot x)]]$$

$$\lesssim \frac{1}{n}\mathbb{E}_{z,z'}[(v\cdot z)^2(v\cdot z')^2(z\cdot z')^{k^\star-2}]$$

$$\lesssim \frac{d^{-(k^\star+2)/2}}{n}$$

In addition, we will also show that the projection term $\mathbb{E}_z[(v\cdot z)^2 y\sigma'(z\cdot x)(z\cdot x)]$ is $O(1/d)$ subexponential. This follows by:

$$\left|\mathbb{E}_z[(v\cdot z)^2 y\sigma'(z\cdot x)(z\cdot x)]\right| \lesssim \left|\mathbb{E}_z[(v\cdot z)^2\sigma'(z\cdot x)(z\cdot x)]\right|$$

$$\lesssim \mathbb{E}_z[(v\cdot z)^2(z\cdot x)^2]$$

$$= \frac{\|x\|^2 + 2(v\cdot x)^2}{d(d+2)}$$

By triangle inequality, this is just $O(1/d)$ subexponential, since the chi-squared $\|x\|^2$ is $O(d)$ subexponential. Therefore, Bernstein's inequality tells us with probability at least $1 - \delta/9^d$:

$$\left\|v^\top\left[\frac{1}{n}\sum_{i=1}^n\mathbb{E}_z[zy_i\sigma'(z\cdot x_i)(x_i\cdot z)z^\top] - \mathbb{E}_{i,z}[zy_i\sigma'(z\cdot x_i)(x_i\cdot z)z^\top]\right]v\right\|$$

$$\lesssim \sqrt{\frac{d^{-(k^\star+2)/2}\log(9^d/\delta)}{n}} + \frac{\log(9^d/\delta)}{dn}$$

Combining the main term and the projection term, we have that for arbitrary $v \in S^{d-1}$, with probability at least $1 - \delta/9^d$:

$$\left\|v^\top\left[\frac{1}{n}\sum_{i=1}^n\mathbb{E}_z[zy_i\sigma'(z\cdot x_i)x_i^\top P_z^\perp] - \mathbb{E}_{i,z}[zy_i\sigma'(z\cdot x_i)x_i^\top P_z^\perp]\right]v\right\|$$

$$\lesssim \sqrt{\frac{d^{-(k^\star+2)/2}\log(9^d/\delta)}{n}} + \frac{\log(9^d/\delta)}{dn}$$

$$= \sqrt{\frac{d^{-(k^\star+2)/2}(d+\log(1/\delta))}{n}} + \frac{d+\log(1/\delta)}{dn}$$

We now consider a $1/4$-net $\mathcal{N}_{1/4}$ over $S^{d-1}$, which has size at most $9^d$. Union bounding over $\mathcal{N}_{1/4}$, we have that with probability at least $1 - \delta$ that:

$$\sup_{v\in\mathcal{N}_{1/4}}\left\|v^\top\left[\frac{1}{n}\sum_{i=1}^n\mathbb{E}_z[zy_i\sigma'(z\cdot x_i)x_i^\top P_z^\perp] - \mathbb{E}_{i,z}[zy_i\sigma'(z\cdot x_i)x_i^\top P_z^\perp]\right]v\right\|$$

$$\lesssim \sqrt{\frac{d^{-(k^\star+2)/2}(d+\log(1/\delta))}{n}} + \frac{d+\log(1/\delta)}{dn}$$

Using the fact that the supremum over the $1/4$-net upper bounds the operator norm up to constant factors, we obtain:

$$\left\|\frac{1}{n}\sum_{i=1}^n\mathbb{E}_z[zy_i\sigma'(z\cdot x_i)x_i^\top P_z^\perp] - \mathbb{E}_{i,z}[zy_i\sigma'(z\cdot x_i)x_i^\top P_z^\perp]\right\|_2$$

$$\lesssim \sqrt{\frac{d^{-(k^\star+2)/2}(d+\log(1/\delta))}{n}} + \frac{d+\log(1/\delta)}{dn}$$

as desired. $\qquad\square$

**Proposition 5.** *When $n \gtrsim d^{k^\star/2}/\Delta^2$ for $\Delta \in (0, 1)$, it holds with probability at least $1 - e^{-d}$ that the top eigenvector $v$ of $\mathbb{E}_z[G]$ satisfies $(v \cdot \theta^\star)^2 \geq 1 - \Delta$.*

*Proof.* This follows directly from the Davis-Kahan theorem since with probability $1 - \delta$ we have:

$$\|\mathbb{E}_z[zb(z)^\top] - \mathbb{E}_{i,z}[zb(z)^\top]\|_2 \lesssim \Delta d^{-k^\star/2}$$

The similar holds true for $\mathbb{E}_z\big[b(z)z^\top\big]$, and hence it holds for the random matrix $G$ as well. Since the eigengap of $\mathbb{E}_i G$ is is $\Theta(d^{-k^\star/2})$, we have the desired result. $\qquad\square$

### H.3 Lipschitzness of $b$

**Lemma 37.** *With probability at least $1 - e^{-cd}$,*

$$\sup_\theta \|b(\theta)\| \lesssim 1 + \sqrt{\frac{d}{n}}.$$

*Proof.* Let $X \in \mathbb{R}^{n \times d}$ be the stacked matrix with all the data points. Then,

$$\|b(\theta)\| = \left\|\frac{1}{n}\sum_{i=1}^n y_i P_\theta^\perp x_i \sigma'(\theta \cdot x_i)\right\| \leq \frac{1}{n}\|X\|_2 \sqrt{\sum_{i=1}^n y_i^2 \sigma'(\theta \cdot x_i)^2} \lesssim 1 + \sqrt{\frac{d}{n}}.$$

$\qquad\square$

**Lemma 38.** *In the same setting as Lemma 37*

$$\sup_\theta \|b(\theta) - b(\theta')\| \leq (1 + \sqrt{d/n})\|\theta - \theta'\|.$$

*Proof.* We have

$$\|b(\theta) - b(\theta')\| \leq \frac{1}{n}\sum_{i=1}^n y_i \big[P_\theta^\perp \sigma'(\theta \cdot x_i) - P_{\theta'}^\perp \sigma'(\theta' \cdot x_i)\big] x_i.$$

Now we have that:

$$P_\theta^\perp \sigma'(\theta \cdot x_i) - P_{\theta'}^\perp \sigma'(\theta' \cdot x_i)$$
$$= P_\theta^\perp [\sigma'(\theta \cdot x_i) - \sigma'(\theta' \cdot x_i)] + \sigma'(\theta' \cdot x_i)[P_\theta^\perp - P_{\theta'}^\perp].$$

For the first term, the same argument as above proves that the sum is bounded by:

$$O\left(\frac{\|X\|_2}{\sqrt{n}}\|\theta - \theta'\|\right) \lesssim (1 + \sqrt{d/n})\|\theta - \theta'\|.$$

For the second term, it is bounded by:

$$O\left(\frac{\|X\|_2 \|P_\theta^\perp - P_{\theta'}^\perp\|_2}{\sqrt{n}}\right) \lesssim (1 + \sqrt{d/n})\|\theta - \theta'\|$$

which completes the proof. $\qquad\square$

