# OpenReview forum: "Improved high-dimensional estimation with Langevin dynamics and stochastic weight averaging"
_ICLR.cc/2026/Conference — ICLR 2026 Poster_

### Official Review · Reviewer_kQAb · 2025-10-16

**Soundness:** 3
**Presentation:** 3
**Contribution:** 2
**Rating:** 6
**Confidence:** 3

**Summary:**

This paper investigates the ability of Langevin dynamics on the sphere to recover a latent signal $\theta^\star$ in the spiked PCA and single-index models. The key idea is to consider the time average of the iterates instead of the final one. The author shows that this averaging procedure allows recovery of $\theta^\star$ with a sample complexity matching that of smoothed online SGD, suggesting that iterate averaging has an implicit smoothing effect. The paper also provides a heuristic connection between mini-batch SGD and Langevin dynamics.

**Strengths:**

The idea that averaging iterates can replace explicit loss smoothing to recover a planted signal with optimal sample complexity in the online regime is novel and conceptually appealing. The paper provides a clear theoretical treatment of this idea in the context of Langevin dynamics on the sphere, supported by sound derivations and proofs.

**Weaknesses:**

(1) The analysis is limited to the continuous-time setting, and it remains unclear how a discretized version of the proposed dynamics would perform in practice or whether it would preserve the same sample complexity guarantees.

(2) While the idea of using iterate averaging could potentially lead to simplified algorithms, the specific procedure analyzed in this work is not practical. It requires oracle knowledge of the information exponent $k^\star$, and depending on its parity, a different estimator must be used.

(3) The use of iterate averaging is quite common in convex optimization, yet this connection to existing literature is not discussed.

(4) The heuristic connection between mini-batch SGD and Langevin dynamics presented in Section~5.2 is somewhat confusing, as it requires $\varepsilon = 1/\eta$ while both quantities are assumed to be small.

(5) The algorithm implemented in the experiments differs from the one analyzed theoretically, and no empirical comparison is provided with smoothing SGD or with the partial-trace method.

**Questions:**

(1) The role of the empirical risk minimization (ERM) formulation (as opposed to an online or streaming setting) is mentioned as a distinction from prior work. However, the algorithm used in experiments appears very close to the standard spherical SGD analyzed in previous works. Is this distinction mostly of conceptual interest? Furthermore, other works such as [1] already provided a similar ``ERM'' formulation, so the novelty of this contribution is not entirely clear to me.

(2) The theoretical analysis is conducted in continuous time, but in practice, the Langevin dynamics must be discretized. The choice of step size can crucially affect both stability and sample complexity, as discussed for instance by Arous et al.(2021). Could the authors comment on how the results would translate to the discrete-time setting? In particular, what scaling of the discretization step would be required for the discrete algorithm to inherit the same statistical guarantees as the continuous process?


[1] Learning Multi-Index Models with Neural Networks via Mean-Field Langevin Dynamics, Mousavi-Hosseini et al. (2025)

---

> ### Author Response · Authors · 2025-11-23
> **Response to Reviewer kQAb (1/2)**
>
> We thank the reviewer for their efforts in providing feedback for our paper. Below, we address the concerns and questions; please let us know if there is anything else we can clarify!
>
> >**Q1: The analysis is limited to the continuous-time setting, and it remains unclear how a discretized version of the proposed dynamics would perform in practice or whether it would preserve the same sample complexity guarantees.**
>
> >**Q5: The algorithm implemented in the experiments differs from the one analyzed theoretically, and no empirical comparison is provided with smoothing SGD or with the partial-trace method.**
>
> A1, A5: We would like to first clarify that our theoretical setting is a heuristic simplification of pure mini-batch SGD, in the continuous time regime. In principle, we can analyze the Euler-Maruyama discretization as well, at the cost of a more tedious analysis. Indeed, since we are not in an online setting, we can take the step size $\eta$ to be arbitrarily small and run for $1/\eta$ time to mitigate any discretization errors. We should also expect that sample complexity will remain the same, since the stationary distribution will be close to the continuous-time estimator, which recovers $\theta^\star$ when $n \gtrsim d^{k^\star/2}$. However, we believe the more interesting technical question is the setting mentioned in our conjecture: proving that pure mini-batch SGD with time averaging also recovers $\theta^\star$, which we feel may require significant more technical work. Nonetheless, our continuous time setting, as is, already circumvents a lower bound on Langevin dynamics by [1], which we believe is sufficiently interesting and surprising.
>
> [1] Ben Arous et al. (2020). Algorithmic thresholds for tensor pca.
>
> >**Q2: While the idea of using iterate averaging could potentially lead to simplified algorithms, the specific procedure analyzed in this work is not practical. It requires oracle knowledge of the information exponent $k^\star$ , and depending on its parity, a different estimator must be used.**
>
> A2: It is true that the estimator we use will depend on the parity of $k^\star$; however, in practice this is not a strong limitation. This is because we can simply run both estimators in parallel, and use a held-out set to determine which one is better. Moreover, the estimator we use in the odd case (direct iterate averaging) is already used in many practical settings as an alternative to learning rate decay.
>
> >**Q3: The use of iterate averaging is quite common in convex optimization, yet this connection to existing literature is not discussed.**
>
> A3: We appreciate this point and agree that the paper currently under-emphasizes the connection to existing literature in iterate averaging. To that end, we will update the related work section to include discussion on this. We do want to emphasize that in our work, the goal of averaging is to smooth out a collection of low-correlation iterates which are all stuck near initialization, whereas the classical setting of Polyak averaging generally revolves around variance reduction near the end of training.
>
> >**Q4: The heuristic connection between mini-batch SGD and Langevin dynamics presented in Section~5.2 is somewhat confusing, as it requires $\epsilon=1/\eta$ while both quantities are assumed to be small.**
>
> A4: We thank the reviewer for pointing this out, since this is indeed a nuanced point. To clarify, $\eta$ is not necessarily assumed to be small. In our experiments, we run with both small and large $\eta$. In the discrete-time setting, small $\eta$ causes SGD to converge to gradient flow which is noiseless and is known to suffer from a sample complexity of $d^{k-1}$ rather than $d^{k/2}$. However, running minibatch SGD with large $\eta$ would boost the noise and allow for fast mixing, and this empirically achieves our conjectured $d^{k/2}$ sample complexity.

---

> ### Author Response · Authors · 2025-11-23
> **Response to Reviewer kQAb (2/2)**
>
> >**Q6: The role of the empirical risk minimization (ERM) formulation (as opposed to an online or streaming setting) is mentioned as a distinction from prior work. However, the algorithm used in experiments appears very close to the standard spherical SGD analyzed in previous works. Is this distinction mostly of conceptual interest? Furthermore, other works such as [1] already provided a similar "ERM" formulation, so the novelty of this contribution is not entirely clear to me.**
>
> A6: We agree that the idea of ERM is definitely not new, but in the context of single index models, prior work such as [2] and [3] indeed uses online SGD to avoid having to track the dependencies introduced by reusing samples across iterates. In contrast, our method reuses  samples (both in Langevin dynamics and mini-batch SGD). For the case of mini-batch SGD in our experiments, this is especially notable because we now have data dependencies as well as a data-dependent stationary distribution, which leads to additional complications for analysis; hence, why we leave this idealized setting as a conjecture.
>
> [1] Mousavi-Hosseini et al. (2025). Learning Multi-Index Models with Neural Networks via Mean-Field Langevin Dynamics.
>
> [2] Damian et al. (2023). Smoothing the landscape boosts the signal for sgd: Optimal sample complexity for learning single index models.
>
> [3] Ben Arous et al. (2021). Online stochastic gradient descent on non-convex losses from high-dimensional inference.
>
> >**Q7: The theoretical analysis is conducted in continuous time, but in practice, the Langevin dynamics must be discretized. The choice of step size can crucially affect both stability and sample complexity, as discussed for instance by Arous et al.(2021). Could the authors comment on how the results would translate to the discrete-time setting? In particular, what scaling of the discretization step would be required for the discrete algorithm to inherit the same statistical guarantees as the continuous process?**
>
> A7: We thank the reviewer for this detailed question. First, we would like to point out that in Ben Arous et al. (2021), the choice of step size is incredibly important because if it is too large, the noise will dominate the drift and you won’t converge, and if it is too small, convergence will take too many steps and as you are in an online SGD setting, this will hurt your sample complexity. On the other hand, because we are in an ERM setting rather than an online setting, the convergence time does not affect our sample complexity so we are safe to take our step size $\eta$ to be arbitrarily small, and run for $1/\eta$ time. This guarantees that we are arbitrarily close to the continuous time Langevin dynamics. More importantly, this will guarantee the same stationary distribution which will allow us to learn with $d^{k/2}$ samples. While we have not derived the precise rate for this Euler-Maruyama discretization, it would be of the order $d^{-O(k)}$, which would give a run-time of $d^{O(k)}$ steps.

---

### Official Review · Reviewer_Vo4P · 2025-10-30

**Soundness:** 4
**Presentation:** 3
**Contribution:** 2
**Rating:** 6
**Confidence:** 4

**Summary:**

The paper shows how combining Langevin dynamics with stochastic weight averaging can enhance signal recovery in high-dimensional estimation tasks by navigating complex loss landscapes more effectively.

**Strengths:**

The paper conveys an interesting check about the possibility to dynamically smooth the landscape in a number of hard inference problems.
It is robust in terms of derivations and conclusions.

**Weaknesses:**

Not sure that it is relevant to Machine Learning applications where landscape is expected to be way smoother already thanks to overparametrization and large datasets.
The paper is not very innovative both concerning ideas and execution. I do not need to specify further the very rich literature on the same subject already cited in the paper and in other parts of this review. Also little parts of the proof are contained in other papers (as explicitly mentioned in the text).
I suggest the authors to stress the added value of their contribution more explicitly, also with reference to possible ML applications especially as they intendo to discuss SGD implications in terms of stochastic averaging.

**Questions:**

Concerning the general idea of dynamical smoothing can you comment on the relation between your proposal and https://www.pnas.org/doi/10.1073/pnas.1608103113 and other papers on the same lines such as https://arxiv.org/abs/1611.01838?
The finite learning rate could represent an effective correlation between the regions over which information is dynamically averaged.
Note that an optimal protocol for the coupling is considered in the contexts of dynamical explorations inspired by robust ensemble.

Possibly related to this discussion: in figure 1 and figure 2 is there a non monotonicity appearing in the behaviour when learning rate increases? Can you discuss why that is? Is there a best learning rate for the smoothing process?
What are the drawbacks deteriorating the quality of the average information for large and small learning rates?

Also a significant chunk of literature about tensor PCA and its sample complexity for the success of Langevin dynamics was overlooked: https://arxiv.org/abs/1812.09066, https://arxiv.org/abs/1902.00139, https://arxiv.org/abs/1907.08226, please consider adding it.

---

> ### Author Response · Authors · 2025-11-23
> **Response to Reviewer Vo4P**
>
> We thank the reviewer for their efforts in providing feedback for our paper, as well as the additional references. Below, we address the concerns and questions; please let us know if there is anything else we can clarify!
>
> >**Q1: Not sure that it is relevant to Machine Learning applications where landscape is expected to be way smoother already thanks to overparametrization and large datasets.**
>
> A1: We agree that in overparameterized neural networks, the training loss landscape is generally benign. Nevertheless, many tasks in the learning pipeline often require nontrivial high-dimensional estimation where the lower order statistics are not informative (e.g. representation learning, self-supervised learning etc.), which is the regime covered by single index models and tensor PCA with high $k$. Moreover, even for such benign landscapes, there can be kernel lower bounds for learnability, and feature learning is the only way to get around them. Our work provides a concrete example for which noisy GD/SGD can help us recover the full signal. We also believe that similar dynamics may arise whenever the population dynamics require escaping a higher-order saddle point, as in sparse parity, modular arithmetic, or multi-step reasoning. Single-index models provide a simple and tractable sandbox for studying these types of dynamics.
>
> >**Q2: Concerning the general idea of dynamical smoothing can you comment on the relation between your proposal and [https://www.pnas.org/doi/10.1073/pnas.1608103113](https://www.pnas.org/doi/10.1073/pnas.1608103113) and other papers on the same lines such as [https://arxiv.org/abs/1611.01838](https://arxiv.org/abs/1611.01838)? The finite learning rate could represent an effective correlation between the regions over which information is dynamically averaged. Note that an optimal protocol for the coupling is considered in the contexts of dynamical explorations inspired by robust ensemble.**
>
> A2: We thank the reviewer for pointing out several references to dynamical smoothing in robust ensemble settings. Our work is similar in spirit, but the mechanism is different - we do not consider a system of interacting replicas; rather, we simply consider a single trajectory of Langevin dynamics and its induced time average. The effective smoothing in our setting comes from the stationary distribution of our noisy dynamics. Actually, the idea of using replicas to smooth the landscape for tensor PCA was also explored in [1], which was a precursor to the landscape smoothing result of [2]. We will add a more thorough discussion of these particle-based methods to a revised version of the paper.
>
> [1] Biroli et al. (2020). How to iron out rough landscapes and get optimal performances: averaged gradient descent and its application to tensor pca.
>
> [2] Damian et al. (2023). Smoothing the landscape boosts the signal for sgd: Optimal sample complexity for learning single index models.
>
> >**Q3: Possibly related to this discussion: in figure 1 and figure 2 is there a non monotonicity appearing in the behaviour when learning rate increases? Can you discuss why that is? Is there a best learning rate for the smoothing process? What are the drawbacks deteriorating the quality of the average information for large and small learning rates?**
>
> A3: We believe there indeed exists a non-monotonicity behavior in the empirical case of mini-batch SGD. The intuition is as following: when the learning rate becomes small, the process approaches gradient flow, and when the learning rate tends to infinity, the stationary distribution approaches the uniform distribution over the $n$ data points (and hence loses all signal) since the projected gradient term is dominated by the $\eta y_i \sigma'(\theta\cdot x_i) x_i$ term. Therefore, we conjecture that for the mini-batch SGD case, there exists a learning rate regime which is large enough to avoid gradient flow, yet small enough to retain signal when we average. We emphasize that this characterization is slightly different from our heuristic connection to Langevin dynamics, in which any sufficiently small $\epsilon$, when run for a sufficiently long time, will have a stationary distribution that recovers the planted direction $\theta^\star$.
>
> >**Q4: Also a significant chunk of literature about tensor PCA and its sample complexity for the success of Langevin dynamics was overlooked: https://arxiv.org/abs/1812.09066, https://arxiv.org/abs/1902.00139, https://arxiv.org/abs/1907.08226, please consider adding it.**
>
> A4: We appreciate the additional references to the extensive literature in spiked matrix-tensor models. Although our setting of tensor PCA and single index models is different from the aforementioned work, they are definitely related in the sense of understanding mechanisms of escaping local minima in non-convex landscapes, as well as the use of Langevin dynamics. We apologize for this oversight, and we plan to include a discussion of these in a revised version of the paper.

---

### Official Review · Reviewer_yNf2 · 2025-11-04

**Soundness:** 3
**Presentation:** 2
**Contribution:** 2
**Rating:** 6
**Confidence:** 3

**Summary:**

The paper proposes to solve the single index models by a Langevin algorithm and demonstrates that this achieves the state of the art theoretical scaling with respect to the information exponent of the link function, on par with smoothing the loss landscape.

**Strengths:**

The paper addresses the problem of learning the implanted direction with a Langevin algorithm, even though this was previously considered inefficient and shows that it is not. The paper nicely explains that it does not solve the problem of escaping from the "equator" for individual samples, but only once the averaging over the samples is performed. It also demonstrates this with numerical simulations.

**Weaknesses:**

It is not clear what are the implications of this work and what should a practitioner gather from these findings.
As far as I understand, Langevin dynamics only boosts the performance when the information exponent k is larger than 2 (otherwise max(1, k-1) = 1 and so is max(1, k/2)). It is not clear to me how relevant is the k > 2 case, especially if the single index model is viewed as a toy model of the neural network. In Example 1, k = 1 and k = 2 cases are often encountered, while cases with k > 2 seem made up.

**Questions:**

1. Compared to the landscape smoothing, what are the pros and cons of using the Langevin algorithm? Some clarification of how the landscape is smoothened would be useful. Can smoothing be applied in practice, if so, could it be shown in Figure 1?

2. What is the practical relevance of the findings in the paper, which practical problems are modeled by the problems considered here (single index models with k > 2 and tensor PCA)? In other words, in which problems should the practitioner consider applying the Langevin algorithm?

3. What is the intuition as to why the Hermite polynomials play an important role here and what is an intuitive meaning of the information exponent?

---

> ### Author Response · Authors · 2025-11-23
> **Response to Reviewer yNf2**
>
> We thank the reviewer for their efforts in providing feedback for our paper. Below, we address the concerns and questions; please let us know if there is anything else we can clarify!
>
> > **Q1: Compared to the landscape smoothing, what are the pros and cons of using the Langevin algorithm? Some clarification of how the landscape is smoothened would be useful. Can smoothing be applied in practice, if so, could it be shown in Figure 1?**
>
> A1: In the landscape smoothing setting of [1], the smoothed loss function at a given $\theta$ is an expectation with respect to the uniform distribution over $S^{d-1}$. Therefore, running that algorithm in practice would require sampling over the sphere to estimate the expectation, which is both expensive and not a natural algorithm. On the other hand, Langevin dynamics on the sphere is simply noisy gradient descent and can be easily implemented in practice. More important is the empirical observation that standard mini-batch SGD, when combined with iterate averaging, indeed recovers our planted direction $\theta^\star$. The Langevin dynamics serves as a theoretical setting in which not only yields a tractable analysis, but also surprisingly circumvents the lower bounds of [2], by the use of time averaging. Whereas in [1] the smoothed loss directly yields a landscape with high signal-to-noise ratio, in our Langevin dynamics/mini-batch SGD setting, we remain in a low signal-to-noise ratio regime, and the time averaging effectively smooths out the noise over time.
>
> [1] Damian et al. (2023). Smoothing the landscape boosts the signal for sgd: Optimal sample complexity for learning single index models.
>
> [2] Ben Arous et al. (2020). Algorithmic thresholds for tensor pca.
>
> > **Q2: What is the practical relevance of the findings in the paper, which practical problems are modeled by the problems considered here (single index models with k > 2 and tensor PCA)? In other words, in which problems should the practitioner consider applying the Langevin algorithm?**
>
> A2: The settings of single index models and tensor PCA that we study serve as sandboxes for classes of problems in which we can hope to unambiguously detect a signal (for instance, in tensor PCA we would like to recover a low-rank structure amongst higher order interactions). From a practitioner's perspective, we believe the takeaway of our work should be conceptual rather than an explicit training recipe - we demonstrate that even in such a noise dominated learning regime, an appropriately tuned noisy gradient descent or mini-batch SGD can exploit higher-order information through some stationary distribution, even when individual iterates seem stuck. From a practical perspective, we believe that similar dynamics may arise whenever the population dynamics require escaping a higher-order saddle point, as in sparse parity, modular arithmetic, or multi-step reasoning. Single-index models provide a simple and tractable sandbox for studying these types of dynamics.
>
> > **Q3: What is the intuition as to why the Hermite polynomials play an important role here and what is an intuitive meaning of the information exponent?**
>
> A3: The Hermite polynomials are the unique orthogonal basis of polynomials for a function under Gaussian inputs. For a link function $\sigma(t)$, we can write its Hermite expansion as $\sigma(t) = \sum_{k=0}^\infty a_k h_k(t)$, where $a_k$ is the $k$th Hermite coefficient, and $h_k$ is the $k$th Hermite polynomial. Then the information exponent is the smallest $k$ with $a_k\neq 0$, which intuitively represents the lowest order moment for which the signal leaves a detectable footprint. As higher-order moments are more expensive to estimate from samples, this will be reflected in the final sample complexity of $d^{k/2}$ which scales with the information exponent $k$.

---

### Meta-Review · Area_Chair_BkZh · 2026-01-06

**Summary:**

This submission is close to the acceptance threshold, and reviewers appreciated the paper’s theoretical contribution, namely showing that Langevin dynamics combined with iterative averaging can match optimal sample complexity for single-index models and tensor PCA, and found the analysis technically sound. At the same time, several reviewers expressed reservations about practical relevance, novelty, and scope, questioning whether the regimes studied arise in realistic machine learning settings, and highlighting gaps between the continuous-time theory and the discretized algorithms used in practice. The rebuttal addressed many of these concerns by clarifying the conceptual takeaways, positioning the results as a theoretical sandbox for understanding noise-dominated learning and higher-order signal recovery, and by situating the work more carefully within the existing literature on landscape smoothing and Langevin dynamics. While the paper remains limited in immediate practical impact, the reviewers’ feedback suggests that the contribution is conceptually meaningful and technically correct, with value as a theoretical insight rather than a ready-to-use algorithm. On balance, and in light of the reviewers’ comments, this supports a weak accept recommendation for a borderline paper whose strengths outweigh its limitations, provided its scope and implications are interpreted appropriately.

**Reviewer Concerns:**

- Reviewer yNf2:
The main concern was the practical relevance of the results, particularly whether the gains for information exponent  k>2 correspond to realistic learning problems beyond stylized single-index models. The reviewer was unconvinced that the  regime is common in practice and felt the implications for practitioners were not sufficiently clear.

- Reviewer Vo4P: The reviewer questioned the novelty and impact of the contribution relative to a rich existing literature on dynamical smoothing, Langevin dynamics, and tensor PCA. They were not fully convinced that the proposed perspective leads to new practical insights for modern machine learning, and suggested that the added value over prior work should be emphasized more clearly.

- Reviewer kQAb: The reviewer raised concerns about the gap between theory and practice, noting that the analysis is restricted to continuous-time Langevin dynamics while experiments rely on discretized SGD-like algorithms. Additional concerns included the practicality of the estimator, which depends on knowledge of the information exponent, and the lack of direct empirical comparisons with smoothing-based baselines.

**Reviewer Scores:**

- Reviewer yNf2: Unchanged, as the rebuttal clarified conceptual takeaways and intuition but did not fundamentally change the reviewer’s view on the limited practical scope.

- Reviewer Vo4P: Unchanged, since the rebuttal addressed literature connections and interpretation but did not alter the reviewer’s assessment of novelty and overall impact.

- Reviewer kQAb: Unchanged, as the rebuttal clarified theoretical intent and limitations but the reviewer’s concerns about practicality and discretization remained.

---

### Decision · Program_Chairs · 2026-01-26

Accept (Poster)